# KromHC: Manifold-Constrained Hyper-Connections with Kronecker-Product Residual Matrices

**Wuyang Zhou** [1]  **Yuxuan Gu** [1]  **Giorgos Iacovides** [1]  **Danilo Mandic** [1]

## Abstract

The success of Hyper-Connections (HC) in neural networks (NN) has also highlighted issues related to training instability and restricted scalability. The Manifold-Constrained Hyper-Connections (mHC) mitigate these challenges by projecting the residual connection space onto a Birkhoff polytope, however, it faces two issues: 1) its iterative Sinkhorn-Knopp (SK) algorithm does not always yield exactly doubly stochastic residual matrices; 2) mHC incurs a prohibitive $\mathcal{O}(n^3C)$ parameter complexity with $n$ as the width of the residual stream and $C$ as the feature dimension. The recently proposed mHC-lite reparametrizes the residual matrix via the Birkhoff-von-Neumann theorem to guarantee double stochasticity, but also faces a factorial explosion in its parameter complexity, $\mathcal{O}\left(nC \cdot n!\right)$. To address both challenges, we propose **KromHC**, which uses the Kronecker products of smaller doubly stochastic matrices to parametrize the residual matrix in mHC. By enforcing manifold constraints across the factor residual matrices along each mode of the tensorized residual stream, KromHC guarantees exact double stochasticity of the residual matrices while reducing parameter complexity to only $\mathcal{O}(n^2C)$. Experiments show that KromHC matches or even outperforms other state-of-the-art (SOTA) mHC variants, while requiring significantly fewer trainable parameters. The code is at `https://github.com/wz1119/KromHC`.

## 1. Introduction

Hyper-Connections (HC) (Zhu et al., 2025) have emerged as a powerful alternative to the ubiquitous residual connections

[1]Department of Electrical and Electronic Engineering, Imperial College London, London, United Kingdom. Correspondence to: Wuyang Zhou <wuyang.zhou19@imperial.ac.uk>.

*Proceedings of the 43$^{rd}$ International Conference on Machine Learning*, Seoul, South Korea. PMLR 306, 2026. Copyright 2026 by the author(s).

*Table 1.* Comparisons between state-of-the-art (SOTA) mHC variants. Our proposed KromHC is the only method that simultaneously achieves *exactly* doubly stochastic residual matrices, parameter efficiency, and requires no specialized kernel optimization.

| Criterion | mHC | mHC-lite | **KromHC (Ours)** |
|---|---|---|---|
| Doubly Stochastic | ⚠️ | ✔ | ✔ |
| Parameter Efficient | ⚠️ | ✘ | ✔ |
| PyTorch Native | ✘ | ✔ | ✔ |

(He et al., 2016). This is achieved by expanding the residual stream width to enhance topological complexity. Unlike the standard residual mapping $\mathbf{x}_{l+1} = \mathbf{x}_l + \mathcal{F}(\mathbf{x}_l)$ (He et al., 2016), HC increases the stream width by an expansion rate, $n$, without incurring computational overhead regarding FLOPs. By introducing learnable mixing across the multiple residual streams, HC allows for more expressive feature propagation. A single layer of HC is defined as

$$\mathbf{X}_{l+1} = \mathbf{H}_l^{\text{res}}\mathbf{X}_l + \mathbf{H}_l^{\text{post}^\top} \mathcal{F}\left(\mathbf{H}_l^{\text{pre}}\mathbf{X}_l\right), \qquad (1)$$

where $\mathbf{X}_l \in \mathbb{R}^{n \times C}$ and $\mathbf{X}_{l+1} \in \mathbb{R}^{n \times C}$ are the expanded input and output at the $l$-th HC layer; $\mathbf{H}_l^{\text{res}} \in \mathbb{R}^{n \times n}$, $\mathbf{H}_l^{\text{pre}} \in \mathbb{R}^{1 \times n}$, and $\mathbf{H}_l^{\text{post}} \in \mathbb{R}^{1 \times n}$ are learnable mappings that, respectively, mix the residual streams, aggregate features from $n$ streams into one, and map the layer output back onto the $n$ streams; $\mathcal{F}(\cdot)$ is a learned residual function such as the attention mechanism (Vaswani et al., 2017).

Recent work (Xie et al., 2025) has suggested that the unconstrained residual matrices ($\{\mathbf{H}_l^{\text{res}}\}_{l=1}^L$) in HC can lead to numerical instabilities when training large-scale neural networks (NNs) such as large language models (LLMs). In particular, HC cannot preserve the identity mapping property of the standard residual connections (He et al., 2016) when stacked across multiple layers, as $\prod_{i=1}^{L-l} \mathbf{H}_{L-i}^{\text{res}}$ fails to preserve the global mean of the features in

$$\mathbf{X}_L = \left(\prod_{i=1}^{L-l} \mathbf{H}_{L-i}^{\text{res}}\right) \mathbf{X}_l$$
$$+ \sum_{i=l}^{L-1} \left(\prod_{j=1}^{L-1-i} \mathbf{H}_{L-j}^{\text{res}}\right) \mathbf{H}_i^{\text{post}^\top} \mathcal{F}(\mathbf{H}_i^{\text{pre}}\mathbf{X}_i), \qquad (2)$$

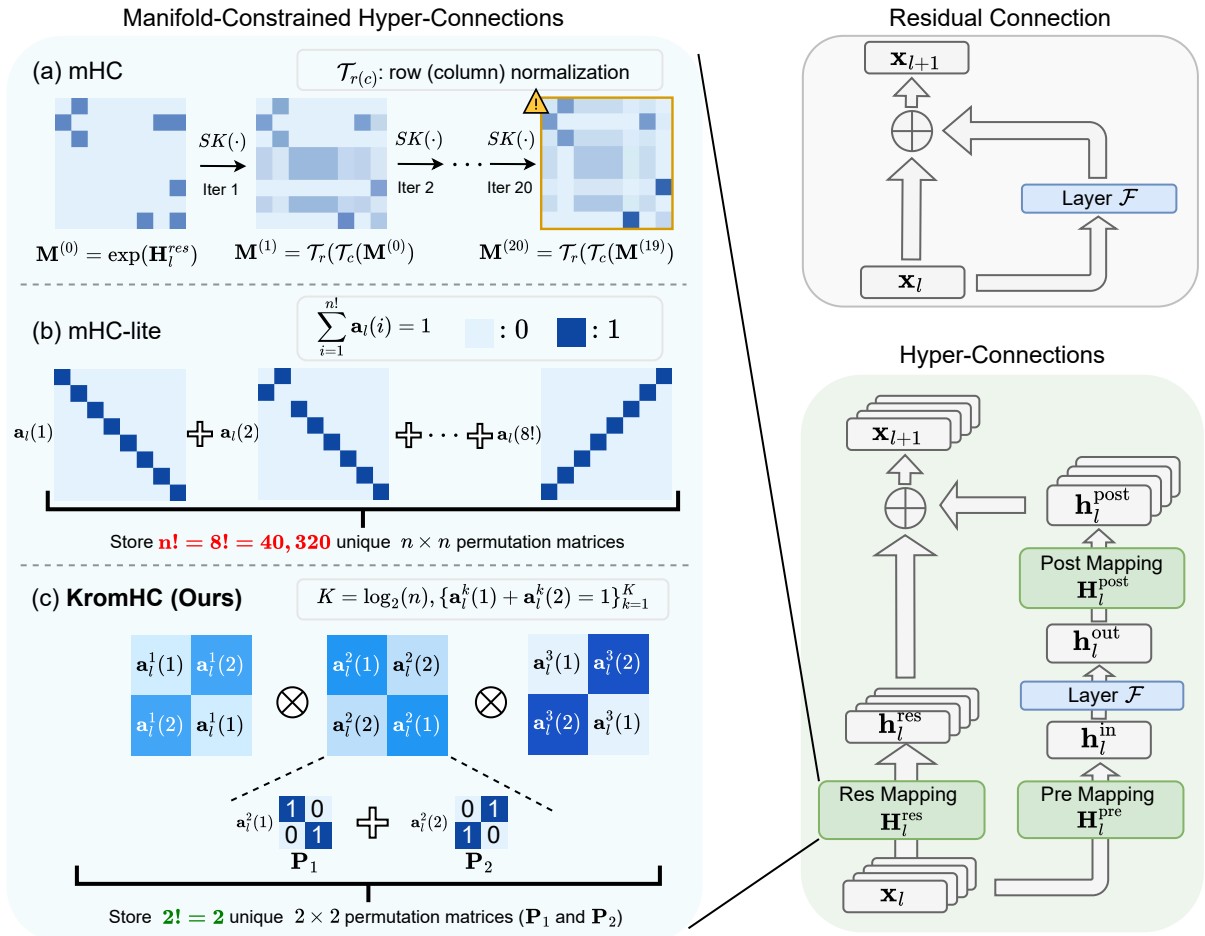

*Figure 1.* Illustration of variants of manifold-constrained hyper-connections with a residual stream width $n = 8$. **(a) mHC**: utilizes iterative Sinkhorn-Knopp (SK) algorithm to approximate a doubly stochastic residual matrix; **(b) mHC-lite**: builds the residual matrix as convex combinations of $n!$ permutation matrices, but becomes infeasible for a large $n$; **(c) KromHC (Ours)**: constructs the residual matrix as the Kronecker products of smaller (e.g., $2 \times 2$) doubly stochastic matrices, thus guaranteeing double stochasticity while remaining parameter efficient.

where $L$ and $l$ represent a deeper and a shallower layer, respectively (Xie et al., 2025).

To address the training instability issue of HC, authors in Xie et al. (2025) have proposed the *Manifold-Constrained Hyper-Connections*, which applies the Sinkhorn-Knopp algorithm (Sinkhorn & Knopp, 1967) to iteratively project the residual matrices, $\{\mathbf{H}_l^{res}\}_{l=1}^L$, onto the Birkhoff polytope (i.e. the set of doubly stochastic matrices). Since the sum of the individual rows and columns of doubly stochastic matrices is always equal to 1, the residual mixing mapping, $\mathbf{H}_l^{res}\mathbf{X}_l$, becomes a convex combination of the input features, preserving feature mean across layers and regularizing the norm of the residual matrices.

However, the Sinkhorn-Knopp algorithm (Sinkhorn & Knopp, 1967) in mHC can fail to achieve double stochasticity when employed over a finite number of iterations (e.g., 20 iterations in mHC). This leads to error accumulation

across layers and undermines training stability (Xie et al., 2025; Yang & Gao, 2026) (See Figure 2). To this end, Yang & Gao (2026) proposed mHC-lite, which guarantees exact double stochasticity by using the *Birkhoff-von-Neumann theorem* (Birkhoff, 1946) to parametrize the residual matrices as the convex combinations of $n \times n$ permutation matrices.

Despite achieving exactly doubly stochastic residual matrices, mHC-lite suffers from an explosion in parameter complexity, as it requires $n!$ unique permutation matrices of size $n \times n$ to be stored. Furthermore, the generic mHC (Xie et al., 2025) has a parameter complexity of $\mathcal{O}(n^3C)$, thus preventing effective scaling of the residual stream width $n$ (See Figure 3). Therefore, the following question naturally arises:

*Can we achieve exact double stochasticity of the residual matrices without incurring an explosion in parameter count as the width of the residual stream, $n$, increases?*

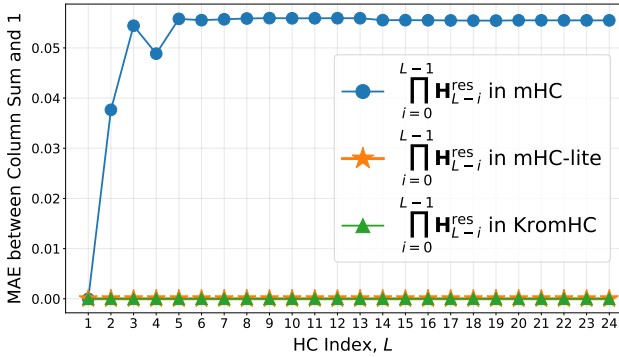

*Figure 2.* Numerical stability analysis of the products of residual matrices. The plot compares the Mean Absolute Error (MAE) between the column sum of $\prod_{i=0}^{L-1} \mathbf{H}_{L-i}^{\text{res}}$ and 1 in an LLM with $D = 12$ transformer blocks and $L = 24$ layers of HC. The standard mHC architecture exhibits a MAE of around 0.05, indicating potential training instabilities. The mHC-lite and KromHC have exactly doubly stochastic matrices, thus yielding zero MAE.

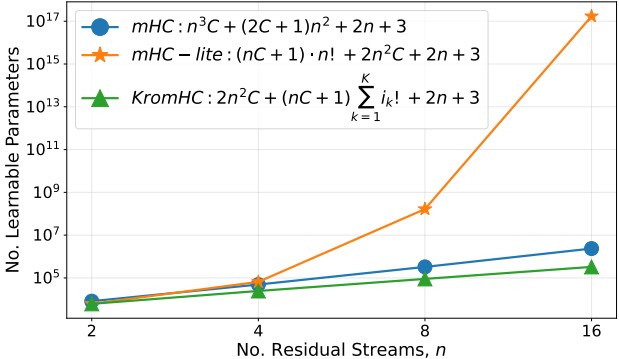

*Figure 3.* The number of learnable parameters against the number of residual streams, $n$, per hyper-connection in mHC, mHC-lite, and KromHC. We assume the feature dimension, $C$, to be 512. Also, $n$ is factored into $\prod_{m=1}^{\log_2(n)} 2$, i.e., $i_1 = i_2 = \cdots = i_K = 2$.

To answer this question, we propose **KromHC**, which uses the Kronecker products (Van Loan, 2000) of smaller doubly stochastic matrices to parametrize the residual matrix in mHC. By framing residual mixing as a Tucker-structured tensor network (Tucker, 1966; Kolda & Bader, 2009; Cichocki et al., 2015) with a core tensor comprising of the tensorized residual streams, we induce a Kronecker structure that guarantees exact double stochasticity of the residual matrices, while having a parameter complexity of $\mathcal{O}\left(n^2 C\right)$. More specifically, KromHC parametrizes the residual matrices, $\{\mathbf{H}_l^{\text{res}}\}_{l=1}^{L}$, as Kronecker products of smaller doubly stochastic matrices, which are learned as convex combinations of smaller permutation matrices as shown in Figure 1. A qualitative comparison between mHC, mHC-lite and the proposed KromHC is shown in Table 1.

In summary, the contributions of this paper are as follows:

- Based on the Kronecker product, we propose KromHC, a novel manifold-constrained hyper-connection framework, providing a link to tensorized residual mixing.

- We resolve the conflict between exact double stochasticity and parameter efficiency in SOTA mHC variants. The proposed KromHC guarantees exactly doubly stochastic residual matrices while enabling more parameter-efficient scaling of the residual stream width.

- We demonstrate the effectiveness and scalability of our approach through extensive experiments on LLM pretraining, achieving consistent improvements over SOTA mHC variants without requiring customized kernels.

## 2. Related Works

**Macro-design of Neural Architecture.** Macro-design concerns the topological structure of blocks in a NN, deciding how inputs and outputs of different blocks are routed and merged across layers (Srivastava et al., 2015). Despite the success of ResNet (He et al., 2016), the use of a single residual stream restricts information flow to a single pathway, which may limit the representational capacity of very deep networks (Zhu et al., 2025). To this end, recent research has focused on expanding the width of the residual stream (Chai et al., 2020; Fang et al., 2023; Mak & Flanigan, 2025; Xiao et al., 2025; Xie et al., 2023; Zhu et al., 2025). For example, Hyper-Connections expands the residual stream into multiple streams and introduces learnable matrices to dynamically mix streams as in Equation (1) (Zhu et al., 2025). However, these methods (Xiao et al., 2025; Mak & Flanigan, 2025; Zhu et al., 2025) may not preserve the identity mapping property of the original residual connection, causing instabilities during training.

**Manifold-Constrained Hyper-Connections.** Based on the original HC (Zhu et al., 2025), DeepSeek recently proposed the Manifold-Constrained Hyper-Connections (Xie et al., 2025). The mHC preserves the identity mapping property of the standard residual connection by projecting the residual matrices, $\{\mathbf{H}_l^{\text{res}} \in \mathbb{R}^{n \times n}\}_{l=1}^{L}$, onto a specific manifold, known as the Birkhoff polytope, $\mathcal{B}_n$. These matrices, $\mathbf{H}_l^{\text{res}}$, are doubly stochastic matrices, which have the following properties

$$\mathbf{H}_l^{\text{res}\top}\mathbf{1}_n = \mathbf{H}_l^{\text{res}}\mathbf{1}_n = \mathbf{1}_n, \mathbf{H}_l^{\text{res}} \geqslant 0, \qquad (3)$$

where $\mathbf{1}_n$ represents an $n$-dimensional vector of all ones, and $\mathbf{H}_l^{\text{res}} \geqslant 0$ means that all entries in $\mathbf{H}_l^{\text{res}}$ are non-negative. Since doubly stochastic matrices have spectral norms equal to 1, and the set is closed under matrix multiplication

(Birkhoff, 1946), this manifold restores the identity mapping property across layers.

Given the input hidden matrix $\mathbf{X}_l \in \mathbb{R}^{n \times C}$ at the $l$-th layer, it is first flattened into a vector $\mathbf{x}_l = \mathrm{vec}(\mathbf{X}_l) \in \mathbb{R}^{1 \times nC}$ to preserve full context information. Then, the learnable residual mappings in mHC are obtained as

$$
\begin{cases}
\mathbf{x}'_l = \mathrm{RMSNorm}(\mathbf{x}_l), \\
\mathbf{H}_l^{\mathrm{pre}} = \sigma\left(\alpha_l^{\mathrm{pre}} \mathbf{x}'_l \mathbf{W}_l^{\mathrm{pre}} + \mathbf{b}_l^{\mathrm{pre}}\right), \\
\mathbf{H}_l^{\mathrm{post}} = 2\sigma\left(\alpha_l^{\mathrm{post}} \mathbf{x}'_l \mathbf{W}_l^{\mathrm{post}} + \mathbf{b}_l^{\mathrm{post}}\right), \\
\mathbf{H}_l^{\mathrm{res}} = \mathrm{SK}\left(\alpha_l^{\mathrm{res}} \cdot \mathrm{mat}\left(\mathbf{x}'_l \mathbf{W}_l^{\mathrm{res}} + \mathbf{b}_l^{\mathrm{res}}\right)\right),
\end{cases}
\tag{4}
$$

where $\mathbf{W}_l^{\mathrm{pre}}, \mathbf{W}_l^{\mathrm{post}} \in \mathbb{R}^{nC \times n}$ and $\mathbf{W}_l^{\mathrm{res}} \in \mathbb{R}^{nC \times n^2}$ are learnable projection matrices; $\mathbf{b}_l^{\mathrm{pre}}, \mathbf{b}_l^{\mathrm{post}} \in \mathbb{R}^{1 \times n}$ and $\mathbf{b}_l^{\mathrm{res}} \in \mathbb{R}^{1 \times n^2}$ are learnable bias terms; the terms $\alpha_l^{\mathrm{pre}}$, $\alpha_l^{\mathrm{post}}$, and $\alpha_l^{\mathrm{res}}$ are learnable scalars, $\mathrm{mat}(\cdot)$ is a reshape function from $\mathbb{R}^{1 \times n^2}$ to $\mathbb{R}^{n \times n}$, and $\sigma(\cdot)$ denotes the Sigmoid function. The $\mathrm{SK}(\cdot)$ operator denotes 20 iterations of the Sinkhorn-Knopp algorithm (Sinkhorn & Knopp, 1967) for projecting the residual matrix onto the Birkhoff polytope. However, mHC does not guarantee double stochasticity and requires customized kernels for accelerating the SK algorithm.

Yang & Gao (2026) proposed mHC-lite to parameterize the doubly stochastic residual matrices as convex combinations of permutation matrices via the Birkhoff-von-Neumann theorem (Birkhoff, 1946) (See Appendix F). It guarantees exact double stochasticity and can be implemented with PyTorch native matrix operations (Paszke et al., 2019). However, the parameter complexity of mHC-lite grows factorially, i.e., $\mathcal{O}(nC \cdot n!)$ with the residual stream width $n$, preventing the scaling of $n$ (See Figure 3).

**Tensor Networks.** Tensor Networks (TNs) provide an efficient representation of higher-order tensors by factorizing them into a network of lower-order cores and factors, thereby alleviating the "curse of dimensionality" (Novikov et al., 2015; Kolda & Bader, 2009; Cichocki et al., 2016; Wang et al., 2023). By exploiting the multi-linear and low-rank structures in NNs, TNs enable expressive yet parameter-efficient representations that can scale efficiently. Recent works have demonstrated their effectiveness in LLM applications such as model compression (Xu et al., 2023; Gu et al., 2025a), parameter-efficient fine-tuning (Bershatsky et al., 2024; Yang et al., 2024; Gu et al., 2025b), etc.

## 3. Notation and Preliminaries

The mathematical notations used in this paper are listed in Table 2. This is consistent with the notation used in Cichocki et al. (2015).

An order-$K$ tensor, $\mathcal{X} \in \mathbb{R}^{i_1 \times i_2 \times \cdots \times i_K}$, is a multi-dimensional array with $K$ modes. A vector is an order-1

*Table 2.* Mathematical notations

| Symbol | Meaning |
|---|---|
| $a, \mathbf{a}, \mathbf{A}, \mathcal{A}$ | Scalar, vector, matrix, tensor |
| $(\cdot)^{\top}$ | Matrix transpose |
| $\mathcal{A}(i_1, \ldots, i_N)$ | The $(i_1, \ldots, i_N)$-th element of $\mathcal{A}$ |
| $\mathbf{I}_{C \times C}$ | Identity matrix of size $C \times C$ |
| $\mathcal{A} \times_n \mathbf{B}$ | Mode-$n$ product |
| $\mathbf{A} \otimes \mathbf{B}$ | Kronecker product |
| $a!$ | $a$ factorial |
| $\mathrm{vec}(\mathcal{A})$ | Vectorization of $\mathcal{A}$ |
| $\mathrm{mat}(\mathcal{A})$ | Matricization of $\mathcal{A}$ |

tensor, and a matrix is an order-2 tensor. Tensorization (folding) reshapes a vector or a matrix into a higher-order tensor. For example, we can tensorize a matrix $\mathbf{A} \in \mathbb{R}^{j_1 \times j_2}$ into an order-$K$ tensor $\mathcal{A} \in \mathbb{R}^{i_1 \times \cdots \times i_K}$, provided that $\prod_{m=1}^{k} i_m = j_1$ and $\prod_{m=k+1}^{K} i_m = j_2$ for some split point $k \in \{1, \ldots, K\}$. The inverse process of tensorization is called unfolding (matricization or vectorization).

**Tucker Decomposition Tensor Network.** Tucker decomposition is a cornerstone in multi-linear tensor networks, as it generalizes the matrix singular value decomposition (SVD) to higher-order tensors (Kolda & Bader, 2009; De Lathauwer et al., 2000). More specifically, Tucker decomposition tensor network parametrizes an order-$K$ tensor, $\mathcal{Y} \in \mathbb{R}^{i_1 \times i_2 \times \cdots \times i_K}$, as

$$
\mathcal{Y} = \mathcal{X} \times_1 \mathbf{U}^1 \times_2 \mathbf{U}^2 \times_3 \cdots \times_K \mathbf{U}^K,
\tag{5}
$$

or in unfolded form

$$
\begin{aligned}
\mathrm{vec}(\mathcal{Y}) &= \left(\mathbf{U}^K \otimes \mathbf{U}^{K-1} \otimes \cdots \otimes \mathbf{U}^1\right) \mathrm{vec}(\mathcal{X}) \\
&= \bigotimes_{k=K}^{1} \mathbf{U}^k \, \mathrm{vec}(\mathcal{X}),
\end{aligned}
\tag{6}
$$

where $\mathcal{X} \in \mathbb{R}^{r_1 \times r_2 \times \cdots \times r_K}$ is an order-$K$ core tensor, $\{\mathbf{U}^k \in \mathbb{R}^{i_k \times r_k}\}_{k=1}^{K}$ are the $K$ factor matrices, and $\mathrm{vec}(\cdot)$ is the operation that converts a tensor from $\mathbb{R}^{i_1 \times i_2 \times \cdots \times i_K}$ to a vector $\in \mathbb{R}^{i_1 \cdot i_2 \cdots i_K}$. The vector containing $[r_1, r_2, \ldots, r_K]$ is the so-called Tucker ranks. The element-wise definition of the mode-$n$ product in $\mathcal{Y} = \mathcal{X} \times_k \mathbf{U}^k$ is

$$
\begin{aligned}
&\mathcal{Y}(r_1, \cdots, r_{k-1}, i_k, r_{k+1}, \cdots, r_K) = \\
&\sum_{r=1}^{r_k} \mathcal{X}(r_1, \cdots, r_{k-1}, r, r_{k+1}, \cdots, r_K) \mathbf{U}^k(i_k, r).
\end{aligned}
\tag{7}
$$

## 4. Methodology

KromHC keeps the parametrization of $\mathbf{H}_l^{\mathrm{post}}$ and $\mathbf{H}_l^{\mathrm{pre}}$ unchanged from mHC, and parametrizes the residual mixing mapping of Equation (1), $\mathbf{H}_l^{\mathrm{res}} \mathbf{X}_l$, as a Tucker decomposition tensor network where the tensorized residual stream is

the core tensor. The proposed KromHC guarantees that all residual matrices are always exact doubly stochastic, while having a learnable parameter count much lower than mHC and mHC-lite. The architecture of the proposed KromHC is illustrated in Figure 1.

## 4.1. Tensorizing the Residual Stream

Let $\mathbf{x}_l \in \mathbb{R}^C$ be the original input feature at the $l$-th layer. We expand the width of the residual stream into $n$, yielding $\mathbf{X}_l \in \mathbb{R}^{n \times C}$ at the $l$-th layer. Given $n = \prod_{k=1}^K i_k, i_k \in \mathbb{Z}^+$, we first tensorize the residual stream into an order-$(K+1)$ tensor, $\mathcal{X}_l \in \mathbb{R}^{i_1 \times i_2 \times \cdots \times i_K \times C}$ (See Figure 7 in Appendix). Afterwards, we perform residual mixing along each of the first $K$ modes of $\mathcal{X}_l$ with the learned doubly stochastic matrices, $\{\mathbf{U}_l^k \in \mathbb{R}^{i_k \times i_k}\}_{k=1}^K$, which satisfy

$$\mathbf{U}_l^k \mathbf{1}_{i_k} = \mathbf{U}_l^{k^\top} \mathbf{1}_{i_k} = \mathbf{1}_{i_k}, \mathbf{U}_l^k \geqslant 0, \text{ for } 1 \leqslant k \leqslant K. \quad (8)$$

This is achieved as

$$\begin{aligned} \mathbf{H}_l^{\text{res}} \mathbf{X}_l = \ & \text{mat}(\mathcal{X}_l \times_1 \mathbf{U}_l^1 \times_2 \mathbf{U}_l^2 \times_3 \\ & \cdots \times_K \mathbf{U}_l^K \times_{K+1} \mathbf{I}_{C \times C}), \end{aligned} \quad (9)$$

where $\text{mat}(\cdot)$ represents the matricization of a tensor from $\mathbb{R}^{i_1 \times i_2 \times \cdots \times i_K \times C}$ to $\mathbb{R}^{n \times C}$. This coincides with the definition of the Tucker decomposition tensor network where the Tucker ranks are equal to the original dimensions, i.e. $[r_1, r_2, \ldots, r_K, r_{K+1}] = [i_1, i_2, \ldots, i_K, C]$, and the last factor matrix, $\mathbf{U}_l^{K+1} \in \mathbb{R}^{C \times C}$, is the identity matrix. Therefore, we can write Equation (9) in the following format

$$\mathbf{H}_l^{\text{res}} \mathbf{X}_l = \underbrace{\left(\mathbf{U}_l^K \otimes \mathbf{U}_l^{K-1} \otimes \cdots \otimes \mathbf{U}_l^1\right)}_{\mathbf{H}_l^{\text{res}}} \mathbf{X}_l, \quad (10)$$

where $\otimes$ denotes the Kronecker product.

Consequently, the single layer propagation in KromHC can be written as

$$\begin{aligned} \mathbf{X}_{l+1} &= \mathbf{H}_l^{\text{res}} \mathbf{X}_l + \mathbf{H}_l^{\text{post}^\top} \mathcal{F}\left(\mathbf{H}_l^{\text{pre}} \mathbf{X}_l\right) \\ &= \bigotimes_{k=K}^1 \mathbf{U}_l^k \mathbf{X}_l + \mathbf{H}_l^{\text{post}^\top} \mathcal{F}\left(\mathbf{H}_l^{\text{pre}} \mathbf{X}_l\right), \end{aligned} \quad (11)$$

where $\mathcal{F}(\cdot)$ denotes a neural network layer which could be an attention mechanism, a feed-forward network (FFN), etc.

## 4.2. Kronecker-Product Residual Matrices

We detail below how to guarantee the double stochasticity of the so obtained $\mathbf{H}_l^{\text{res}}$ from the Kronecker product of smaller doubly stochastic matrices, $\mathbf{U}_l^k \in \mathbb{R}^{i_k \times i_k}$.

**Theorem 4.1.** (*Birkhoff-von-Neumann Theorem (Birkhoff, 1946)*) *For any $n \times n$ doubly stochastic matrix, $\mathbf{X}$, there*

*exists a finite collection of permutation matrices $\{\mathbf{P}_k \in \mathbb{R}^{n \times n}\}_{k=1}^{n!}$ and a coefficient vector $\mathbf{a} = (a_1, \ldots, a_{n!}) \in \mathbb{R}^{n!}$ satisfying $a_k \geqslant 0, \forall k \in [n!]$ and $\sum_{k=1}^{n!} a_k = 1$, such that $\mathbf{X} = \sum_{k=1}^{n!} a_k \mathbf{P}_k$.*

Since the size of doubly stochastic matrices, $\mathbf{U}_l^k \in \mathbb{R}^{i_k \times i_k}$, are typically much smaller than $n \times n$, we can parametrize them as convex combinations of permutation matrices of shape $i_k \times i_k$ via Theorem 4.1. For example, let the width of the residual stream, $n$, be a power of 2 (i.e., $2, 4, 8, 16, \ldots$) and $\{i_k = 2\}_{k=1}^K$, where $K = \log_{i_k=2}(n)$. In this case, only 2 permutation matrices of size $2 \times 2$ need to be stored for parametrizing all $K$ different $\mathbf{U}_l^k \in \mathbb{R}^{i_k \times i_k}$. Furthermore, we only need to learn 2 scalars in order to represent any $\mathbf{U}_l^k \in \mathbb{R}^{i_k \times i_k}$ on the Birkhoff polytope as the convex combination of two $2 \times 2$ permutation matrices.

**Theorem 4.2.** (*Kronecker Closure of Doubly Stochastic Matrices*) *Let $\mathcal{B}_n \subset \mathbb{R}^{n \times n}$ denote the set of $n \times n$ doubly stochastic matrices. Let $\mathbf{U}_l^1 \in \mathcal{B}_{i_1}$ and $\mathbf{U}_l^2 \in \mathcal{B}_{i_2}$. Then their Kronecker product satisfies*

$$\mathbf{U}_l^1 \otimes \mathbf{U}_l^2 \in \mathcal{B}_{i_1 i_2}. \quad (12)$$

*More generally, for any finite collection $\{\mathbf{U}_k \in \mathcal{B}_{i_k}\}_{k=1}^K$, their iterated Kronecker product satisfies*

$$\bigotimes_{k=K}^1 \mathbf{U}_k \in \mathcal{B}_n, \text{ where } n = \prod_{k=1}^K i_k. \quad (13)$$

*Proof.* See Appendix B or Taranenko (2023). $\square$

Theorem 4.2 states that the Kronecker product of any finite collection of doubly stochastic matrices is also doubly stochastic. Since $\{\mathbf{U}_l^k \in \mathbb{R}^{i_k \times i_k}\}_{k=1}^K$ are doubly stochastic matrices and $\mathbf{H}_l^{\text{res}} = \bigotimes_{k=K}^1 \mathbf{U}_l^k$, $\mathbf{H}_l^{\text{res}}$ in the proposed KromHC is guaranteed to be doubly stochastic. This is equivalent to imposing a Kronecker structure to the residual matrix, $\mathbf{H}_l^{\text{res}}$, which acts as an extra constraint besides the manifold constraint used in mHC.

*Remark* 4.3. Due to the guaranteed double stochasticity of the residual matrices, KromHC preserves the desired properties of the original mHC, such as *norm preservation* and *compositional closure*. Norm preservation means the spectral norm of the residual matrix is bounded by 1 (i.e., $\|\mathbf{H}_l^{\text{res}}\| \leqslant 1$). Compositional closure preserves stability throughout all layers, as $\prod_{i=1}^L \mathbf{H}_{L-i}^{\text{res}}$ remains exact doubly stochastic.

## 4.3. Parametrization of KromHC

We detail below the parametrization of $\mathbf{H}_l^{\text{pre}}$, $\mathbf{H}_l^{\text{post}}$ and $\mathbf{H}_l^{\text{res}}$ in KromHC. We follow mHC to flatten the input, $\mathbf{X}_l \in$

*Table 3.* Comparisons of additional learnable parameters relative to standard residual connections, training loss, validation bits-per-byte (BPB), and CORE score across different types of manifold-constrained hyper connections. The number of transformer blocks is denoted by $D$. Each transformer block has 2 residual connections. All experiments are conducted with $n = 4$ residual streams. The best and second best values among different methods are highlighted in **bold** and underlined, respectively.

| Method | $D = 6$ | | | | $D = 12$ | | | |
|---|---|---|---|---|---|---|---|---|
| | $\Delta$ Params (K) $\downarrow$ | Train Loss $\downarrow$ | Val BPB $\downarrow$ | CORE Score$\uparrow$ | $\Delta$ Params (K) $\downarrow$ | Train Loss $\downarrow$ | Val BPB $\downarrow$ | CORE Score$\uparrow$ |
| Residual | – | 3.490 | 1.047 | 6.477 | – | 2.971 | 0.864 | 14.774 |
| mHC | 462 | 3.493 | **1.042** | 7.971 | 1844 | **2.964** | **0.861** | 16.023 |
| mHC-lite | 609 | **3.484** | 1.045 | 8.208 | 2433 | 2.972 | 0.864 | 13.217 |
| KromHC (**Ours**) | **240** | 3.488 | 1.047 | **9.018** | **959** | 2.966 | 0.862 | **16.872** |

$\mathbb{R}^{n \times C}$, at the $l$-th layer to $\mathbf{x}_l \in \mathbb{R}^{1 \times nC}$. The parametrization of KromHC is as follows:

$$
\begin{cases}
\mathbf{x}'_l = \text{RMSNorm}(\mathbf{x}_l), \\
\mathbf{H}_l^{\text{pre}} = \sigma\left(\alpha_l^{\text{pre}}\mathbf{x}'_l\mathbf{W}_l^{\text{pre}} + \mathbf{b}_l^{\text{pre}}\right), \\
\mathbf{H}_l^{\text{post}} = 2\sigma\left(\alpha_l^{\text{post}}\mathbf{x}'_l\mathbf{W}_l^{\text{post}} + \mathbf{b}_l^{\text{post}}\right), \\
\mathbf{a}_l^k = \text{Softmax}(\alpha_l^{\text{res}}\mathbf{x}'_l\mathbf{W}_l^{\text{res},k} + \mathbf{b}_l^{\text{res},k}), \\
\mathbf{U}_l^k = \sum_{m=1}^{i_k!} \mathbf{a}_l^k(m)\mathbf{P}_m, \\
\mathbf{H}_l^{\text{res}} = \bigotimes_{k=K}^{1} \mathbf{U}_l^k,
\end{cases} \quad (14)
$$

where $n$ is factorized into $K$ terms, i.e., $n = \prod_{k=1}^{K} i_k$, $i_k \in \mathbb{Z}^+$. The term $\mathbf{P}_m \in \mathbb{R}^{n \times n}$ denotes the $m$-th unique permutation matrix. The scalar $\mathbf{a}_l^k(m)$ is the $m$-th entry in $\mathbf{a}_l^k$. The operator $\bigotimes_{k=K}^{1}$ denotes the sequence of Kronecker products. The $a_l^{\text{pre}}$, $a_l^{\text{post}}$, and $a_l^{\text{res}}$ are the learnable scalar coefficients at the $l$-th KromHC layer. The sigmoid function is denoted as $\sigma(\cdot)$. The $\mathbf{W}_l^{\text{pre}} \in \mathbb{R}^{nC \times n}$, $\mathbf{W}_l^{\text{post}} \in \mathbb{R}^{nC \times n}$, and $\mathbf{W}_l^{\text{res},k} \in \mathbb{R}^{nC \times i_k!}$ are the learnable weight matrices. The $\mathbf{b}_l^{\text{pre}} \in \mathbb{R}^{1 \times n}$, $\mathbf{b}_l^{\text{post}} \in \mathbb{R}^{1 \times n}$, and $\mathbf{b}_l^{\text{res},k} \in \mathbb{R}^{1 \times i_k!}$ are the learnable biases.

*Remark* 4.4. Although the set of permutation matrices $\{\mathbf{P}_m\}_{m=1}^{i_k!}$ for each factor $\mathbf{U}_l^k$ is given by construction, the coefficients $\mathbf{a}_l^k$ are learned via Equation (14). Thus, each factor $\mathbf{U}_l^k$ is a learnable point within the Birkhoff polytope $\mathcal{B}_{i_k}$, allowing $\mathbf{H}_l^{\text{res}}$ to dynamically mix the residual streams while maintaining its doubly stochastic property.

*Remark* 4.5. The dimensions $\{i_k\}_{k=1}^K$ can be any integer factorization of $n$ such that $i_k \geqslant 2$. Although any valid factorization preserves the doubly stochastic property of $\mathbf{H}_l^{\text{res}}$, choosing the prime factorization (e.g., $i_k = 2$ for $n = 2^K$) yields the highest parameter efficiency. This is because the number of learnable parameters in KromHC scales with $\sum i_k!$.

**Parameter Complexity Analysis.** The learnable parameter count of KromHC per HC layer is much lower than that of the mHC and mHC-lite (See Figure 3). More specifically,

the parameter complexity of mHC is $\mathcal{O}\left(n^3 C\right)$, while for mHC-lite, the parameter complexity is $\mathcal{O}\left(nC \cdot n!\right)$. The parameter count of KromHC is $2n^2C + (nC+1)\sum_{k=1}^{K} i_k! + 2n + 3$. Since $i_k$ is usually very small (e.g., $i_k = 2$), the parameter complexity of KromHC is dominated by $\mathcal{O}(n^2 C)$.

# 5. Experiments

The evaluation of the training and downstream performance of the proposed KromHC on LLM pretraining was performed on two scales: $\sim 60$M parameters ($D = 6$ transformer blocks) and $\sim 186$M parameters ($D = 12$ transformer blocks) by replacing the residual connections in Nanochat (Karpathy, 2025) (See Section G for more details). All models were trained on the `FineWeb-Edu` (Penedo et al., 2024) dataset with a Token:Parameter ratio of $\sim 20$ following Hoffmann et al. (2022). Experiments were conducted using either 4 or 8 NVIDIA RTX PRO 6000 GPUs, depending on the number of residual streams. Experimental results demonstrate that KromHC matches or outperforms SOTA mHC variants, while using significantly fewer trainable parameters.

## 5.1. Initialization

Following the experimental settings in Yang & Gao (2026), we initialized $\mathbf{W}_l^{\text{res},k}$, $\mathbf{W}_l^{\text{pre}}$ and $\mathbf{W}_l^{\text{post}}$ to zero. The bias vectors $\mathbf{b}_l^{\text{pre}}$ and $\mathbf{b}_l^{\text{post}}$ were set to -1 for all entries except for a single index in each vector, which was set to 1. We set $\alpha_l^{\text{pre}}$ and $\alpha_l^{\text{post}}$ to 0.01.

For $\{i_k = 2\}_{k=1}^K$, there are exactly two permutation matrices: $\mathbf{P}_1 = \begin{bmatrix} 1 & 0 \\ 0 & 1 \end{bmatrix}$ and $\mathbf{P}_2 = \begin{bmatrix} 0 & 1 \\ 1 & 0 \end{bmatrix}$. The $\mathbf{b}_l^{\text{res},k}$ was set to $[0, -8]^T$, and $\alpha_l^{\text{res}}$ was set to 0.01. This initialization ensures that, at initialization, $\mathbf{a}_l^k(1) \approx 1$ and $\mathbf{a}_l^k(2) \approx 0$, yielding $\mathbf{U}_l^k \approx \mathbf{I}_{2 \times 2}$. Consequently, the Kronecker products of identity matrices produced a nearly identity matrix $\mathbf{H}_l^{\text{res}}$ at initialization.

*Table 4.* Commonsense and reasoning benchmark results (accuracy %). $D = 6$ or $12$ transformer blocks and $n = 4$ residual streams were used for the experiments. The best and second best values among different methods are highlighted in **bold** and underlined, respectively.

| $D$ | Method | Avg | HS | HS-ZS | PIQA | ARC-E | ARC-C | COPA | CSQA | OBQA | WG | WGrande | BoolQ |
|---|---|---|---|---|---|---|---|---|---|---|---|---|---|
| 6 | Residual | 39.8 | 27.0 | **29.0** | 60.4 | **42.8** | 21.8 | 55.0 | 31.8 | 23.8 | **59.3** | 49.2 | 37.4 |
| | mHC | 40.8 | 26.4 | 28.4 | **60.8** | 42.8 | 22.4 | **60.0** | 23.0 | 24.8 | 55.3 | **50.4** | 54.4 |
| | mHC-lite | 41.0 | 27.4 | 28.2 | 56.4 | 42.8 | **24.4** | 53.0 | **35.0** | 25.6 | 52.4 | 50.0 | 54.4 |
| | KromHC (**Ours**) | **41.1** | **28.0** | 27.8 | 59.0 | 41.4 | 23.0 | 54.0 | 28.8 | 25.4 | 53.5 | 50.0 | **60.8** |
| 12 | Residual | 46.2 | **36.6** | 35.4 | **65.4** | 58.4 | 26.4 | 63.0 | **36.6** | 32.6 | 57.5 | 52.0 | 44.0 |
| | mHC | 47.5 | 36.2 | 37.6 | 64.2 | **58.6** | 27.4 | 64.0 | 32.4 | **33.6** | 60.4 | 53.2 | 54.6 |
| | mHC-lite | 44.4 | 35.2 | 36.2 | 64.6 | **58.6** | 27.2 | 63.0 | 23.6 | 30.4 | 58.6 | 49.2 | 41.6 |
| | KromHC (**Ours**) | **47.7** | 36.4 | **38.4** | 65.0 | 57.8 | **27.6** | **66.0** | 30.0 | 31.2 | 58.6 | **54.8** | 58.4 |

*Table 5.* Language modeling, BigBench (BBH) subtasks, and evaluation suite results (accuracy %). $D = 6$ or $12$ transformer blocks and $n = 4$ residual streams were used for the experiments. The best and second best values among different methods are highlighted in **bold** and underlined, respectively.

| $D$ | Method | Avg | Lamb | SQuAD | CoQA | BBH-QA | BBH-CS | BBH-Op | BBH-Dyck | LSAT | LangID |
|---|---|---|---|---|---|---|---|---|---|---|---|
| 6 | Residual | 16.2 | 18.6 | 0.0 | 5.6 | **20.0** | 40.8 | **9.1** | 4.6 | 23.5 | 23.4 |
| | mHC | 15.4 | 17.4 | 0.0 | 4.8 | 10.6 | **42.0** | 5.2 | 9.2 | 23.5 | 26.0 |
| | mHC-lite | 16.9 | **19.6** | **0.4** | 4.6 | 19.4 | 38.2 | 5.7 | 6.4 | **29.6** | **28.0** |
| | KromHC (**Ours**) | **17.3** | 19.0 | 0.2 | **5.8** | 14.0 | 40.8 | 8.6 | **11.4** | 27.8 | 27.8 |
| 12 | Residual | 23.7 | 29.2 | **10.8** | 13.8 | 39.2 | 38.8 | 12.9 | **15.8** | 27.8 | 25.4 |
| | mHC | 22.9 | **31.6** | 5.8 | 13.0 | 39.6 | 42.0 | **16.7** | 13.0 | 20.4 | 24.0 |
| | mHC-lite | 23.3 | 30.0 | 8.4 | 14.2 | 36.6 | 42.6 | 14.8 | 10.0 | 27.0 | **26.2** |
| | KromHC (**Ours**) | **24.0** | 30.4 | 8.2 | **15.4** | **40.4** | **44.6** | 11.9 | 13.6 | 26.1 | 25.0 |

## 5.2. Training and Validation Set Metrics

We compared the performances of standard residual connection, mHC, mHC-lite and our proposed KromHC methods under both 6 and 12 transformer blocks with $n = 4$ residual streams. The results are shown in Table 3. The training loss denotes the cross-entropy (CE) loss $\mathcal{L}_{CE} = -\frac{1}{T}\sum_{t=1}^{T}\log p_\theta(x_t|x_{<t})$, while the validation performance is measured using a tokenizer-invariant metric, bits-per-bytes (BPB) (i.e., $\mathcal{L}_{BPB} = \frac{\mathcal{L}_{CE}}{\ln(2)} \times \frac{\text{Total Tokens}}{\text{Total Bytes}}$). CORE score (Li et al., 2024) is the centered accuracy computed over a fixed subset of 22 downstream evaluation tasks to reflect general language understanding quality (See more details in Appendix C).

Notably, our method significantly outperformed SOTA mHC variants in terms of the CORE score, indicating that the models trained with KromHC have stronger capabilities in downstream tasks including commonsense reasoning and language modeling. Additionally, our method achieved on-par training loss and validation BPB as mHC and mHC-lite while having much fewer additional learnable parameters compared to mHC and mHC-lite.

## 5.3. Downstream Task Performances

Table 4 and 5 present the detailed performance evaluations for commonsense reasoning and language modeling, respectively. We compared the proposed KromHC with the residual connection, mHC and mHC-lite. All models were trained under identical settings with 6 or 12 transformer blocks and $n = 4$ residual streams.

**Commonsense Reasoning.** Our method achieved the highest average accuracies at both 6-block ($41.1\%$) and 12-block ($47.7\%$) settings, consistently outperforming standard residual connections and other SOTA manifold-constrained HC variants. In particular, KromHC demonstrates strong capabilities in reasoning-intensive tasks such as COPA and BoolQ, surpassing the second best scores by up to $2\%$ and $6.4\%$ respectively. The consistent improvements across both model depths suggest that KromHC scales effectively with depth, while remaining robust across diverse commonsense reasoning tasks. These results demonstrate that KromHC is beneficial for reasoning tasks.

**Language Modeling.** KromHC also achieved the best average performance ($17.3\%$ and $24.0\%$) in language modeling at $D = 6$ and $D = 12$. These results suggest that KromHC is effective for improving the language modeling performance in LLM pretraining, which is essential for language understanding.

## 5.4. Scaling the Width of Residual Stream in KromHC

In order to assess how the performance of KromHC scales with the width of the residual stream $n$, we conducted experiments on LLM pretraining with 12 transformer blocks and

*Table 6.* Additional number of parameters of our KromHC models with different widths of residual streams compared to the standard residual connection under $D = 12$.

| | $n = 4$ | $n = 8$ | $n = 16$ |
|---|---|---|---|
| $\Delta$ Params (M) | 0.96 | 2.67 | 11.37 |

$n \in \{4, 8, 16\}$ residual stream width. As shown in Figure 4 (left), the gap between training losses becomes larger as $n$ increases. A similar scaling trend is observed in validation, where BPB consistently improves as $n$ increases (See Figure 4 (right)). The additional number of learnable parameters at different $n$ are recorded in Table 6. These results demonstrate that KromHC benefits from larger residual stream width and scales effectively with $n$.

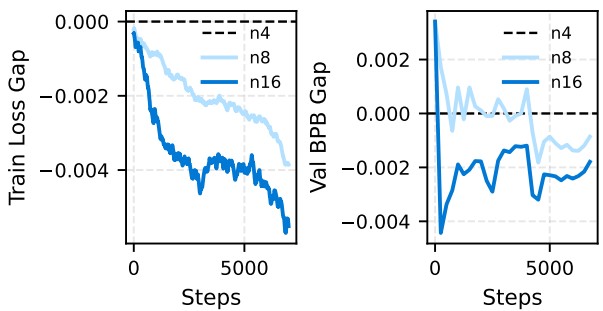

*Figure 4.* Training loss and validation BPB gaps of KromHC at different widths of the residual stream, $n$, compared to $n = 4$. Exponential Moving Average (EMA) is applied to the raw loss before the calculation of the loss gap.

### 5.5. Gradient Norm

Figure 5 presents the gradient norm trajectories across the last 2000 training steps. Identical model configurations (12 transformer blocks and $n = 4$ residual streams) were used for mHC, mHC-lite and our KromHC. It is worth noting that our KromHC consistently achieved the lowest gradient norm compared with other manifold-constrained hyper-connection variants. Both mHC-lite and KromHC achieved lower gradient norms than mHC due to their exactly doubly stochastic residual matrices (See Figure 2). This indicates improved training stability in KromHC and suggests that KromHC can control gradient magnitudes more effectively during training.

### 5.6. Ablation Study

**Shared $\alpha_l^{\text{res}}$.** We examined whether the scaling factor $\alpha_l^{\text{res}}$ should be shared across all $\mathbf{U}_l^k$ or be unique for each matrix, i.e. $\alpha_l^{\text{res}}$ versus $\alpha_l^{\text{res},k}$. In Equation (14), the mixing

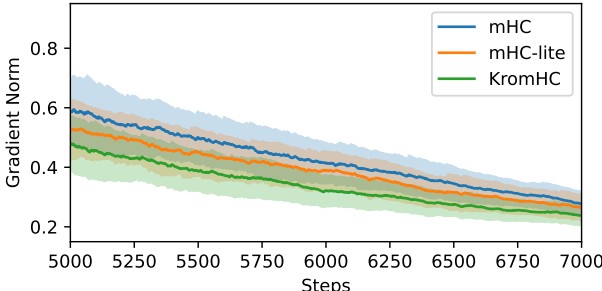

*Figure 5.* Zoomed-in view of gradient norms from 5000 to 7000 steps during training. Trajectories are smoothed using EMA, with shaded regions indicating the EMA variance.

coefficients for permutation matrices were computed as

$$\mathbf{a}_l^k = \text{Softmax}(\alpha_l^{\text{res}} \mathbf{x}_l' \mathbf{W}_l^{\text{res},k} + \mathbf{b}_l^{\text{res},k}). \quad (15)$$

As shown in Figure 6, sharing $\alpha_l^{\text{res}}$ across all $\mathbf{U}_l^k$ yields better performance than learning unique $\alpha_l^{\text{res},k}$ for each matrix.

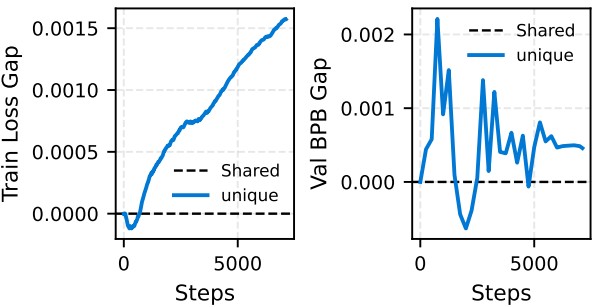

*Figure 6.* Sharing $\alpha_l^{\text{res}}$ across all doubly stochastic matrices $\mathbf{U}_l^k$ outperforms the use of matrix-specific $\alpha_l^{\text{res},k}$. The experiment was conducted with 12 transformer blocks and $n = 4$ residual streams.

## 6. Conclusion

We have introduced KromHC, a parameter-efficient manifold-constrained hyper-connection framework which employs Kronecker-product residual matrices to guarantee their exact double stochasticity. In this way, KromHC also resolves the scalability limitations of existing mHC variants regarding parameter complexity. Extensive experiments have demonstrated the effectiveness of the proposed method in LLM pre-training. Our future work aims to apply KromHC to other domains such as computer vision.

**Limitations.** KromHC may encounter parameter issues when the width of the residual stream $n$ is a large prime number. However, this can be mitigated by using a larger $n$ which is a power of 2 or 3 or exhibits a prime factorization consisting of small numbers.

## Impact Statement

This work resolves the scalability and stability issues of manifold-constrained hyper-connections which advances the field of Machine Learning. By enabling more reliable training with fewer parameters, the proposed KromHC supports more accessible and sustainable future deployment of advanced AI systems.

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

## A. Kronecker Product

Kronecker products provide a convenient way to represent structured linear operators. For matrices $\mathbf{A} \in \mathbb{R}^{m \times n}$ and $\mathbf{B} \in \mathbb{R}^{p \times q}$, their Kronecker product $\mathbf{A} \otimes \mathbf{B} \in \mathbb{R}^{(mp) \times (nq)}$ is defined as

$$\mathbf{A} \otimes \mathbf{B} = \begin{bmatrix} a_{11}\mathbf{B} & \cdots & a_{1n}\mathbf{B} \\ \vdots & \ddots & \vdots \\ a_{m1}\mathbf{B} & \cdots & a_{mn}\mathbf{B} \end{bmatrix}, \qquad (16)$$

where each entry of $\mathbf{A}$ scales the entire matrix $\mathbf{B}$. For example, a Kronecker product between two $2 \times 2$ matrices yields a $4 \times 4$ matrix. Let

$$\mathbf{A} = \begin{bmatrix} 1 & 2 \\ 3 & 4 \end{bmatrix}, \qquad \mathbf{B} = \begin{bmatrix} 0 & 5 \\ 6 & 7 \end{bmatrix}.$$

The Kronecker product $\mathbf{A} \otimes \mathbf{B} \in \mathbb{R}^{4 \times 4}$ is

$$\mathbf{A} \otimes \mathbf{B} = \begin{bmatrix} 1\mathbf{B} & 2\mathbf{B} \\ 3\mathbf{B} & 4\mathbf{B} \end{bmatrix} = \begin{bmatrix} 0 & 5 & 0 & 10 \\ 6 & 7 & 12 & 14 \\ 0 & 15 & 0 & 20 \\ 18 & 21 & 24 & 28 \end{bmatrix}. \quad (17)$$

The Kronecker product naturally extends to multiple matrices. Let $\mathbf{A}^k \in \mathbb{R}^{m_k \times n_k}$ for $k = 1, \dots, K$. Their Kronecker product is defined recursively as

$$\bigotimes_{k=K}^{1} \mathbf{A}^i = \mathbf{A}^K \otimes \mathbf{A}^{K-1} \otimes \cdots \otimes \mathbf{A}^1, \qquad (18)$$

and the resulting matrix has size $\left(\prod_{k=1}^{K} m_k\right) \times \left(\prod_{k=1}^{K} n_k\right)$.

## B. Proof for Theorem B.1

**Theorem B.1.** *(**Kronecker Closure of Doubly Stochastic Matrices**) Let $\mathcal{B}_n \subset \mathbb{R}^{n \times n}$ denote the set of $n \times n$ doubly stochastic matrices. Let $\mathbf{U}_l^1 \in \mathcal{B}_{i_1}$ and $\mathbf{U}_l^2 \in \mathcal{B}_{i_2}$. Then their Kronecker product satisfies*

$$\mathbf{U}_l^1 \otimes \mathbf{U}_l^2 \in \mathcal{B}_{i_1 i_2}. \qquad (19)$$

*More generally, for any finite collection $\{\mathbf{U}_k \in \mathcal{B}_{i_k}\}_{k=1}^{K}$, their iterated Kronecker product satisfies*

$$\bigotimes_{k=K}^{1} \mathbf{U}_k \in \mathcal{B}_n, \text{ where } n = \prod_{k=1}^{K} i_k. \qquad (20)$$

*Proof.* For any doubly stochastic matrix, its elements are non-negative, and its row and column sums are equal to one. Let $\mathbf{U}_l^1 \in \mathcal{B}_{i_1}$ and $\mathbf{U}_l^2 \in \mathcal{B}_{i_2}$. The Kronecker product $\mathbf{U}_l^1 \otimes \mathbf{U}_l^2$ yields elements of $\mathbf{U}_l^1(i, j)\mathbf{U}_l^2(k, l)$. The product

of two non-negative real number is non-negative. Therefore, $\mathbf{U}_l^1 \otimes \mathbf{U}_l^2 \geq 0$.

Let $\mathbf{1}_{i_1}$ and $\mathbf{1}_{i_2}$ denote all-one column vectors of dimensions $i_1$ and $i_2$, respectively. We use the Kronecker product identity

$$(\mathbf{A} \otimes \mathbf{B})(\mathbf{C} \otimes \mathbf{D}) = (\mathbf{AC}) \otimes (\mathbf{BD}), \qquad (21)$$

to obtain

$$\begin{aligned}(\mathbf{U}_l^1 \otimes \mathbf{U}_l^2)(\mathbf{1}_{i_1} \otimes \mathbf{1}_{i_2}) &= (\mathbf{U}_l^1 \mathbf{1}_{i_1}) \otimes (\mathbf{U}_l^2 \mathbf{1}_{i_2}) \\ &= \mathbf{1}_{i_1} \otimes \mathbf{1}_{i_2} = \mathbf{1}_{i_1 i_2}.\end{aligned} \qquad (22)$$

Therefore, all row sums of $\mathbf{U}_l^1 \otimes \mathbf{U}_l^2$ equal 1.

Similarly,

$$\begin{aligned}(\mathbf{U}_l^1 \otimes \mathbf{U}_l^2)^\top(\mathbf{1}_{i_1} \otimes \mathbf{1}_{i_2}) &= (\mathbf{U}_l^{1\top} \mathbf{1}_{i_1}) \otimes (\mathbf{U}_l^{2\top} \mathbf{1}_{i_2}) \\ &= \mathbf{1}_{i_1 i_2}.\end{aligned} \qquad (23)$$

Therefore, all column sums of $\mathbf{U}_l^1 \otimes \mathbf{U}_l^2$ equal 1.

Combining the non-negative property with row and column sums of 1, we have showed that $\mathbf{U}_l^1 \otimes \mathbf{U}_l^2$ is doubly stochastic (i.e., $\mathbf{U}_l^1 \otimes \mathbf{U}_l^2 \in \mathcal{B}_{i_1 i_2}$).

By induction, this result extends to a finite collection of $\{\mathbf{U}_k \in \mathcal{B}_{i_k}\}_{k=1}^{K}$, as follows

$$\bigotimes_{k=K}^{1} \mathbf{U}_k \in \mathcal{B}_n, \text{ where } n = \prod_{k=1}^{K} i_k. \qquad (24)$$

$\square$

## C. Details of the CORE Tasks

Authors in (Li et al., 2024) proposed the CORE metric to provide a robust low-variance, centered accuracy score for LLM evaluation. There are 22 selected tasks, where in each task, accuracy is linearly scaled so that 0 indicates random-guess performance and 1 implies perfect accuracy. The final CORE score is averaged across all 22 tasks, preventing any single benchmark dominating the calculation.

The tasks in CORE experiments span logical reasoning, factual recall, algorithmic thinking, commonsense inference, and language understanding. In particular, it includes reasoning and knowledge tasks (Zhong et al., 2024; Clark et al., 2018), BIG-Bench tasks (Srivastava et al., 2023), Question Answering and Commonsense (Clark et al., 2019; Talmor et al., 2019; Gordon et al., 2012), and other widely used benchmarks (Zellers et al., 2019; Kocijan et al., 2023; Sakaguchi et al., 2021).

## D. Tensor Network Diagram of KromHC

Figure 7 shows the TN diagram of our KromHC method. In a tensor network (TN) diagram, a tensor is denoted by a

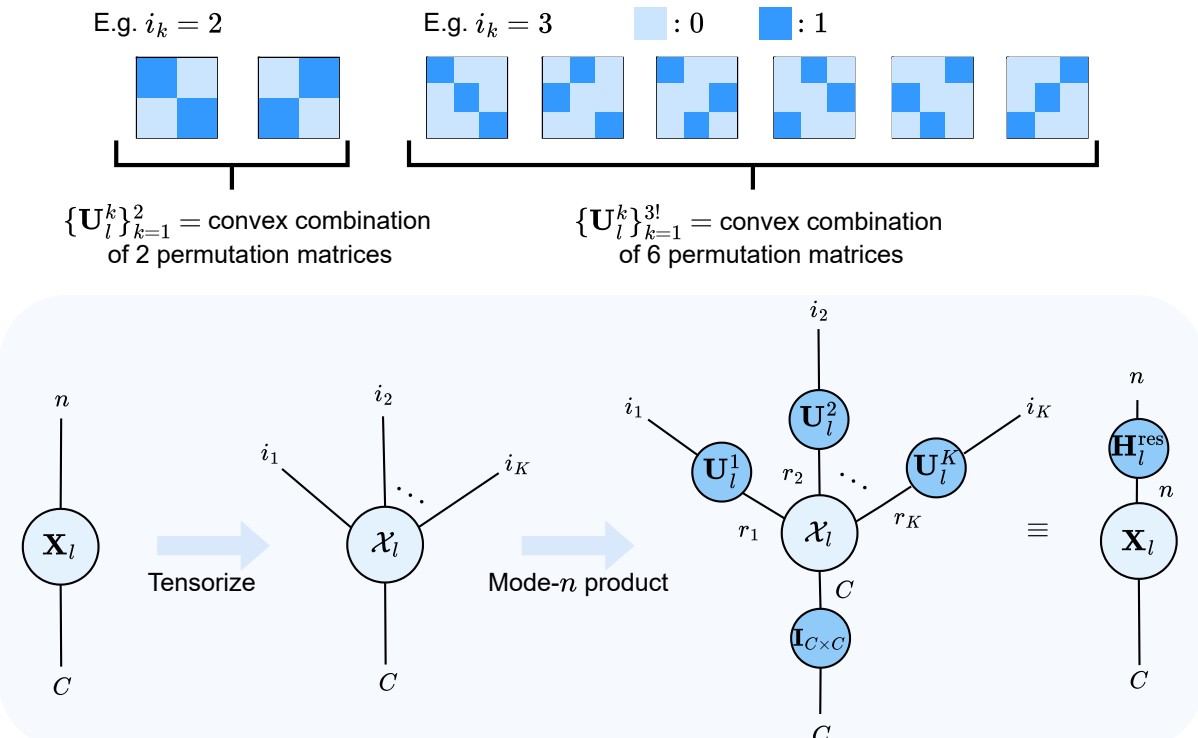

Figure 7. Tensor network diagram of the proposed KromHC method.

circle, with each line emanating from the circle corresponding to a tensor mode index. Also, connecting two index lines implies a tensor contraction over the connected mode indices.

## E. Parametrization of HC

In this section, we detail the parametrization of HC when it was first proposed by Zhu et al. (2025). Given the input hidden matrix $\mathbf{X}_l \in \mathbb{R}^{n \times C}$ at the $l$-th layer, the dynamic mappings and the static mappings are obtained as

$$\begin{cases} \mathbf{X}'_l = \mathrm{RMSNorm}(\mathbf{X}_l), \\ \mathbf{H}^{\mathrm{pre}}_l = \alpha^{\mathrm{pre}}_l \cdot \tanh\left(\mathbf{W}^{\mathrm{pre}}_l \mathbf{X}'^T_l\right) + \mathbf{b}^{\mathrm{pre}}_l, \\ \mathbf{H}^{\mathrm{post}}_l = \alpha^{\mathrm{post}}_l \cdot \tanh\left(\mathbf{W}^{\mathrm{post}}_l \mathbf{X}'^T_l\right) + \mathbf{b}^{\mathrm{post}}_l, \\ \mathbf{H}^{\mathrm{res}}_l = \alpha^{\mathrm{res}}_l \cdot \tanh\left(\mathbf{W}^{\mathrm{res}}_l \mathbf{X}'^T_l\right) + \mathbf{b}^{\mathrm{res}}_l, \end{cases} \quad (25)$$

where $\mathbf{W}^{\mathrm{pre}}_l, \mathbf{W}^{\mathrm{post}}_l \in \mathbb{R}^{1 \times C}$ and $\mathbf{W}^{\mathrm{res}}_l \in \mathbb{R}^{n \times C}$ are linear projections for dynamic mappings, $\mathbf{b}^{\mathrm{pre}}_l, \mathbf{b}^{\mathrm{post}}_l \in \mathbb{R}^{1 \times n}$ and $\mathbf{b}^{\mathrm{res}}_l \in \mathbb{R}^{n \times n}$ are learnable bias terms, and $\mathrm{RMSNorm}(\cdot)$ normalizes the feature dimension $C$. Xie et al. (2025) have discovered that HC can be unstable to train. Also, this formalization of HC does not allow for cross-stream input-dependent residual mixing like in mHC, which may limit its expressivity. Therefore, a better way to implement HC is to keep its parameterization similar to mHC and not use the

Sinkhorn-Knopp algorithm, as their functional difference is the manifold constraint.

## F. Parametrization of mHC-lite

In this section, we detail the parametrization of mHC-lite (Yang & Gao, 2026). Let $\mathbf{X}_l \in \mathbb{R}^{n \times C}$ denote the input feature in the $l$-th layer and $\mathbf{x}_l \in \mathbb{R}^{1 \times nC}$ denote the flattened input feature. Then we build mappings $\mathbf{H}^{\mathrm{res}}_l, \mathbf{H}^{\mathrm{pre}}_l$, and $\mathbf{H}^{\mathrm{post}}_l$ dynamically based on $\mathbf{x}_l$ as

$$\begin{cases} \mathbf{x}'_l = \mathrm{RMSNorm}(\mathbf{x}_l), \\ \mathbf{H}^{\mathrm{pre}}_l = \mathrm{sigmoid}\left(\alpha^{\mathrm{pre}}_l \mathbf{x}'_l \mathbf{W}^{\mathrm{pre}}_l + \mathbf{b}^{\mathrm{pre}}_l\right), \\ \mathbf{H}^{\mathrm{post}}_l = 2 \cdot \mathrm{sigmoid}\left(\alpha^{\mathrm{post}}_l \mathbf{x}'_l \mathbf{W}^{\mathrm{post}}_l + \mathbf{b}^{\mathrm{post}}_l\right), \\ \mathbf{a}_l = \mathrm{softmax}(\alpha^{\mathrm{res}}_l \mathbf{x}'_l \mathbf{W}^{\mathrm{res}}_l + \mathbf{b}^{\mathrm{res}}_l), \\ \mathbf{H}^{\mathrm{res}}_l = \sum_{k=1}^{n!} \mathbf{a}_l(k) \mathbf{P}_k, \end{cases} \quad (26)$$

where $\mathbf{W}^{\mathrm{pre}}_l, \mathbf{W}^{\mathrm{post}}_l \in \mathbb{R}^{nC \times n}$ and $\mathbf{W}^{\mathrm{res}}_l \in \mathbb{R}^{nC \times n!}$ are learnable weight matrices in the $l$-th layer. Here, $\mathbf{b}^{\mathrm{pre}}_l, \mathbf{b}^{\mathrm{post}}_l \in \mathbb{R}^{1 \times n}$ and $\mathbf{b}^{\mathrm{res}}_l \in \mathbb{R}^{1 \times n!}$ are learnable bias terms. The terms $\alpha^{\mathrm{pre}}_l, \alpha^{\mathrm{post}}_l$, and $\alpha^{\mathrm{res}}_l$ are learnable scalars. $\mathbf{P}_m \in \mathbb{R}^{n \times n}$ are permutation matrices.

## G. Nanochat

Each transformer block uses two residual connections, one for the attention mechanism and one for the FFN. Besides the standard residual connections $\mathbf{x}_{l+1} = \mathbf{x}_l + \mathcal{F}(\mathbf{x}_l)$, Nanochat (Karpathy, 2025) introduces learnable per-layer scalars which improves the model performance. More specifically, each layer's input is calcualted as $\tilde{\mathbf{x}}_l = \lambda_l^{\text{resid}} \cdot \mathbf{x}_l + \lambda_l^{x_0} \cdot \mathbf{x}_0$, where $\mathbf{x}_0$ is the initial embedding and $\lambda_l^{\text{resid}}, \lambda_l^{x_0}$ are learnable scalars initialized to 1 and 0 respectively (Jordan, 2024; Wang et al., 2024). These were all replaced with the mHC variants when examining the performance of mHC variants.

## H. Hyperparameters

Table 7 lists the hyperparameters used in our experiments. Note that Muon optimizer (Liu et al., 2025) is used for learning parameters in the main branch including attention and multi-layer perceptron (MLP) weight matrices, while AdamW optimizer (Loshchilov & Hutter, 2019) is used for the hyper-connections streams, embedding layer, and language modeling (LM) head layer. Additionally, following the best practice in Karpathy (2025), different learning rates (LR) are used for embedding layer, LM layer, main branch and hyper-connections branch. For the two different model

*Table 7.* Hyperparameters used in our experiments.

| Name | Value |
| --- | --- |
| Batch size | 524,288 |
| Sequence length | 2048 |
| Head dimension | 128 |
| % Steps for LR warmdown | 0.4 |
| LR for main branch (Muon) | 0.02 |
| Weight decay of Muon optimizer | 0.2 |
| LR for residual connections (AdamW) | 0.005 |
| AdamW $\beta_1$ | 0.8 |
| AdamW $\beta_2$ | 0.95 |

scales, i.e. $D = 6$ and $D = 12$ transformer blocks, the corresponding scale-specific hyperparameters are listed in Table 8.

*Table 8.* Scale-specific hyperparameters used in our experiments for $n = 4$ residual streams.

| Name | $D = 6$ | $D = 12$ |
| --- | --- | --- |
| Hidden dimension | 384 | 768 |
| LR for embedding params (AdamW) | 0.43 | 0.3 |
| LR for LM head params (AdamW) | 0.0057 | 0.004 |
| # Training steps | 2500 | 7000 |

## I. Grad Norm

Figure 8 shows the raw gradient norm across 7000 training steps of mHC, mHC-lite and KromHC at $D = 12$.

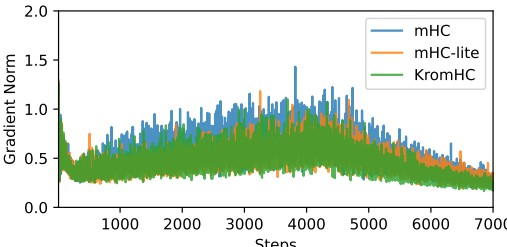

*Figure 8.* Gradient norm dynamics across training. Raw gradient norm across 7000 training steps.

## J. System Metrics

In Table 9, we report the system efficiency of different manifold-constrained hyper-connections ($n = 4$) relative to mHC. These were implemented in pure PyTorch using 12 transformer blocks and evaluated on 8 RTX 6000 GPUs. KromHC is more computationally efficient than mHC and mHC-lite in terms of higher throughput, lower wall-clock per step and a lower GPU memory footprint. Notice that we implement the mHC baseline purely based on PyTorch and may underestimate the metrics of the specialized-kernel implementation in Xie et al. (2025).

*Table 9.* Relative percentage change in the system metrics (token throughput, wall-clock per step, and GPU memory footprint) compared to mHC. "-" indicates no difference.

| Method | Throughput $\Delta$ (%) ↑ | Wall-clock $\Delta$ (%) ↓ | Memory $\Delta$ (%) ↓ |
| --- | --- | --- | --- |
| mHC | - | - | - |
| mHC-lite | +9.39 | -8.86 | -1.04 |
| **KromHC** | **+19.23** | **-16.24** | **-1.15** |

## K. Factorization Choices

To empirically examine whether more restricted factorizations will lead to worse performance, we conducted an ablation study at $n = 8$ with 12 transformer blocks and different factorizations. The original mHC-lite configuration at $n = 8$ is unrealistic due to model size explosion. Comparing the $4 \times 2$ with the more restricted $2 \times 2 \times 2$ factorization in KromHC, the $2 \times 2 \times 2$ achieves slightly better results, achieving a train loss 0.002 lower than $4 \times 2$ and a validation BPB 0.003 lower than $4 \times 2$. This confirms that KromHC resolves the exactness-efficiency trade-off without sacrificing model performance.

