# OpenReview forum: "KromHC: Manifold-Constrained Hyper-Connections with Kronecker-Product Residual Matrices"
_ICML.cc/2026/Conference — ICML 2026 regular_

### Official Review · Reviewer_B8hQ · 2026-03-12

**Soundness:** 3
**Presentation:** 3
**Significance:** 2
**Originality:** 3
**Overall Recommendation:** 4
**Confidence:** 3

**Summary:**

This work introduces KromHC, an evolution of manifold-constrained hyper-connections (mHC) designed to streamline LLM training. The authors propose representing the residual mixing matrix as a Kronecker product of smaller doubly stochastic matrices—each parameterized via a convex combination of permutation matrices. This formulation is particularly clever as it preserves the exact doubly stochasticity of mHC-lite while circumventing its prohibitive factorial parameter growth. Beyond the theoretical guarantees provided, the paper demonstrates through small-scale experiments that KromHC achieves competitive training stability and performance with a significantly reduced parameter footprint.

**Compliance With Llm Reviewing Policy:**

Affirmed.

**Key Questions For Authors:**

1、KromHC restricts the residual mixing matrix to the subclass of Kronecker-structured doubly stochastic matrices. How restrictive is this parameterization in practice?

Since the claimed benefit of KromHC comes from the “exactness + efficiency” trade-off, it would be valuable to quantify the cost of this additional structural constraint. In particular, can the authors provide either a theoretical discussion or a controlled empirical study over different factorizations / factor sizes to show when this restriction is benign and when it becomes limiting?

2、Can the authors provide stronger evidence for the paper’s scalability claim by comparing KromHC against mHC / mHC-lite at larger residual-stream widths (e.g., n>4) or on larger models?

If the authors could directly verify these claims on a larger-scale models, I would be much more inclined to raise my overall recommendation.

3、The paper emphasizes parameter efficiency and theoretical FLOP reductions,, but does not provide direct system-level efficiency such as throughput metrics(tokens/sec), GPU memory, or inference latency. Since specialized kernels are not utilized, it is essential to demonstrate that the proposed tensorization doesn't introduce computational bottlenecks that offset the parameter savings. Providing these metrics would significantly clarify the method's practical utility.

**Limitations:**

Yes

**Strengths And Weaknesses:**

Soundness: The paper's core construction is technically solid. The exact doubly stochasticity of the Kronecker-product residual matrices follows directly from the parameterization and Theorem 4.2. The complexity analysis matches the implementation.

Presentation: The manuscript is well-presented with a clear narrative throughout. And the appendix provides details on hyperparameters, initialization, and implementation, ensuring the experimental framework is accessible.

Significance: The study is well-motivated, addressing the fundamental challenges of stability and scalability in residual routing. However, the current evidence is primarily derived from small-scale models, a single backbone, and relatively limited head-to-head settings. Further validation across varied architectures and larger settings is highly encouraged to fully substantiate the generalizability and robustness of KromHC.

Originality: The paper introduces a nontrivial Kronecker-structured reparameterization of manifold-constrained hyper-connections, linking exact doubly stochastic residual mixing with tensorized residual streams. However, the work is more of a clever structural synthesis of existing ideas than a fundamentally new paradigm, so I view the originality as good.

---

> ### Author Rebuttal · Authors · 2026-03-28
>
> Thank you very much for your positive assessment of the soundness, presentation, significance and originality of our paper. To show the scalability of KromHC to larger models, we report below its performance against HC, mHC and mHC-lite on pre-training an LLM of ~1.08 billion parameters.
>
> ---
>
> > **Q1**. KromHC restricts the residual mixing matrix to the subclass of Kronecker-structured doubly stochastic matrices... Since the claimed benefit of KromHC comes from the “exactness + efficiency” trade-off, it would be valuable to quantify the cost of this additional structural constraint. In particular, can the authors provide either a theoretical discussion or a controlled empirical study over different factorizations / factor sizes to show when this restriction is benign and when it becomes limiting?
>
> **Response**: Thank you for this insightful question. Although the Kronecker structure spans a restricted subset of the Birkhoff polytope, we observe empirically that this constraint acts as a **beneficial regularizer**. Moreover, although mHC-lite spans the whole Birkhoff polytope, it has a **factorial** parameter growth with respect to $n$ and is infeasible at large $n$ (e.g., mHC-lite with 12 transformer blocks and $n=8$ yields ~6 billion parameters, making the hyper-connection parameters account for ~**97%** of the total parameters). Resolving this parameter explosion is a primary contribution of KromHC.
>
> To empirically show that the Kronecker-structure does not degrade performance, we conducted an ablation at $n=8$ with 12 transformer blocks, comparing the more restricted $2 \times 2 \times 2$ against a $4 \times 2$ factorization. The $2 \times 2 \times 2$ setup achieved better train loss and validation BPB. This confirms that the exactness-efficiency trade-off is resolved in KromHC without sacrificing model performance.
>
> |Factorization|Train loss $\downarrow$|Validation BPB $\downarrow$|
> |-|-|-|
> |mHC-lite $n=8$|Model Size Explosion|Model Size Explosion|
> |KromHC $4 \times 2$|2.961|0.863|
> |KromHC $2 \times 2 \times 2$| **2.959**|**0.860**|
>
> ---
>
> > **Q2 & Larger-scale Models.** However, the current evidence is primarily derived from small-scale models... Further validation across larger settings is highly encouraged to fully substantiate the generalizability and robustness of KromHC...  If the authors could directly verify these claims on a larger-scale models, I would be much more inclined to raise my overall recommendation.
>
> **Response**: Thank you very much for the potential in raising the score. We have further scaled the NanoChat model to a larger-scale setting of **~1.08 billion trainable parameters** (26 layers of transformer blocks, Tokens:Params ratio of ~7, and $n=4$). Comparing to HC, mHC, and mHC-lite, KromHC achieves the **best** train loss (2.539) and validation BPB (0.751) and the *second-best* CORE score (0.260) while being the **most efficient** in system metrics such as throughput (150,649 tokens/sec), wall-clock per step (3.499 second per step), and GPU memory footprint (86.91GB per GPU). This result highlights the scalability and system efficiency of the proposed KromHC.
>
> |Method|CORE score $\uparrow$|Train loss $\downarrow$|Validation BPB $\downarrow$|Throughput (tokens/sec) $\uparrow$|Wall-clock per step (second) $\downarrow$|Memory on Each GPU (GB) $\downarrow$|
> |-|-|-|-|-|-|-|
> |HC|0.240|2.554|0.756|133,726|3.932|*87.09*|
> |mHC|**0.262**|**2.539**|**0.751**|137,521|3.883|87.62|
> |mHC-lite|0.250|2.551|0.755|*138,361*|*3.807*|87.20|
> |**KromHC (Ours)**|*0.260*|**2.539**|**0.751**|**150,649**|**3.499**|**86.91**|
>
> ---
>
> > **Q3.** The paper emphasizes parameter efficiency and theoretical FLOP reductions, but does not provide direct system-level efficiency such as throughput metrics(tokens/sec), GPU memory, or inference latency... Providing these metrics would significantly clarify the method's practical utility.
>
> **Response**: The PyTorch-native KromHC is indeed **more efficient** than mHC and mHC-lite on system metrics such as throughput, wall-clock per step, and memory footprint. The response to Q2 above demonstrates the system-level efficiency of KromHC on LLM pretraining of ~1.08 billion trainable parameters. We also report below the system efficiency at 12 transformer blocks and $n=4$ with 8 RTX PRO 6000 GPUs. KromHC is more computationally efficient than mHC and mHC-lite in terms of **higher** throughput (663,730 tokens/sec), **lower** wall-clock per step (0.794 second) and a **lower** GPU memory footprint (68.47 GB per GPU).
>
> |Method|Throughput (tokens/sec) $\uparrow$|Wall-clock per step (second) $\downarrow$|Memory on Each GPU (GB) $\downarrow$|
> |-|-|-|-|
> |mHC|556,689|0.948|69.27|
> |mHC-lite|608,973|0.864|68.55|
> |**KromHC (Ours)**|**663,730**|**0.794**|**68.47**|
>
> ---
>
> We hope that our answers have addressed your concerns. We would like to kindly ask you to consider increasing your score. If any questions remain, we are happy to address them.

---

> > ### Author Rebuttal · Reviewer_B8hQ · 2026-04-05
> >
> > Thanks for the response. My concerns have been adequately addressed.

---

> > > ### Author Response · Authors · 2026-04-05
> > >
> > > Thank you very much for the insightful discussion. We are glad that your concerns have been addressed.

---

### Official Review · Reviewer_E9S1 · 2026-03-12

**Soundness:** 3
**Presentation:** 3
**Significance:** 2
**Originality:** 2
**Overall Recommendation:** 5
**Confidence:** 3

**Summary:**

This paper proposes a specific manifold Hyper-connection, KromHC, through tensorizing the residual stream. By stacking 2x2 permutation matrices to construct tensors, the new architecture ensures exact doubly stochastic residual matrices while enabling more parameter-efficient scaling of the residual stream width. Empirically, KromHC nearly halves the number of mHC parameters, which can be considered a solid improvement.

**Compliance With Llm Reviewing Policy:**

Affirmed.

**Final Justification:**

I thank the authors for providing additional experiments, which gave me a clearer understanding of how KromHC and mHC approximate doubly stochastic matrices. The proposed KromHC outperforms both mHC and mHC-lite in terms of throughput, per-step training time, and GPU memory usage, while also significantly reducing the number of parameters. However, I believe the paper still has some limitations. In particular, applying KromHC requires factorizing the integer n; when multiple factorizations are possible, it remains unclear which one yields the best performance. This makes the proposed method difficult to generalize.

**Key Questions For Authors:**

**Q1**: I am confused by the exclamation mark in Table 1. Does it indicate a state between "yes" and "no"? The paper does not provide an explanation for this symbol. Additionally, I believe it is inappropriate to include "Parameter Efficient" as a criterion, because the main objective of this paper is to introduce a new mHC framework to guarantee doubly stochastic matrices, rather than to compress the original model.

**Q2**: Since KromHC is constructed as a $k$-order tensor composed of matrices of the same size, e.g., $k=log_2(n)$ does this mean there will be a gap when n is not a power of 2? How is this gap addressed in the experiments?

**Q3**: Figure 2 demonstrates the improvement of mHC-lite and KromHC over mHC in approximating doubly stochastic matrices. However, the gap shown for mHC does not seem very obvious to me. Could you calculate the distance between mHC's $\prod_{i=0}^{L-1} H_{L-i}^{res}$ and the Birkhoff polytope?

**Q4**: In the experimental section, we can see that mHC still achieves good results in some cases. The biggest advantage of KromHC by comparison is that it uses fewer parameters. Does this indicate that the performance improvement brought by using a more accurate doubly stochastic matrices structure is limited?

**Limitations:**

yes

**Strengths And Weaknesses:**

Overall, this is a logically clear and rigorously presented work. The proposed structure naturally and reasonably combines manifold HC with tensor networks; however, it lacks significant innovation and deeper insights. Furthermore, its simplification of the mHC parameters theoretically diminishes the model's learning capacity.

---

> ### Author Rebuttal · Authors · 2026-03-28
>
> Thank you for your positive assessment of the rigor and parameter-efficiency of KromHC. We believe there may have been some misunderstanding. Please find our responses below.
>
> ---
>
> > **Q1.** I am confused by the exclamation mark in Table 1... Additionally, I believe it is inappropriate to include 'Parameter Efficient' as a criterion, because the main objective of this paper is to introduce a new mHC framework to guarantee doubly stochastic matrices, rather than to compress the original model.
>
> **Response**: The exclamation mark in Table 1 indicates that mHC does not achieve exact double stochasticity (due to finite Sinkhorn-Knopp iterations) and incurs a prohibitive $\mathcal{O}(n^3 C)$ parameter complexity. We will clarify this in the camera-ready version.
>
> Parameter efficiency is a primary motivation alongside exact double stochasticity. mHC-lite achieves exact double stochasticity but suffers from a **factorial** parameter explosion $\mathcal{O}(nC \cdot n!)$, making it infeasible at large $n$: with 12 transformer blocks and $n=8$, mHC-lite already needs ~6 billion parameters, making the hyper-connection parameters account for ~**97%** of the total parameters. Resolving this parameter explosion is a primary contribution of KromHC.
>
> ---
>
> > **Q2.** Since KromHC is constructed as a k-order tensor composed of matrices of the same size, e.g., k=log_2(n) does this mean there will be a gap when n is not a power of 2? How is this gap addressed in the experiments?
>
> **Response**: There is no gap when $n$ is not a power of 2. As detailed in Lines 230–232 (left column), Section 4.1, and Remark 4.5, any integer factorization of $n$ is valid. For example, if $n=6$, we can construct the residual matrix via the Kronecker product of a $2\times 2$ and $3\times 3$ doubly stochastic matrix. As noted in the limitations section, if $n$ is a prime number, KromHC may have many parameters, but this can be easily mitigated by choosing a slightly larger $n$ with a factorization of small numbers.
>
> ---
>
> > **Q3**. Figure 2 demonstrates the improvement of mHC-lite and KromHC over mHC in approximating doubly stochastic matrices... Could you calculate the distance between mHC's $\sum_{i=0}^{L-1}$ $H^{res}_{L-i} $ and the Birkhoff polytope?
>
> **Response**: We report below the Frobenius norm distance between $\sum_{i=0}^{L-1}$ $H^{res}_{L-i} $ in mHC ($n=4$) and the Birkhoff polytope. It can be seen that the distance increases to ~0.1 after 8 mHC connections. mHC-lite and KromHC maintain a distance of 0.
>
> |Connection|1|2|3|4|5|6|7|8|9|10|11|12|13|14|15|16|17|18|19|20|21|22|23|24|
> |-|-|-|-|-|-|-|-|-|-|-|-|-|-|-|-|-|-|-|-|-|-|-|-|-|
> |mHC|0.000|0.025|0.167|0.064|0.074|0.084|0.095|0.097|0.099|0.100|0.101|0.102|0.103|0.101|0.101|0.100|0.100|0.100|0.100|0.100|0.099|0.099|0.099|0.099|
>
> ---
>
> > **Q4 & W1**. KromHC's simplification of the mHC parameters theoretically diminishes the model's learning capacity... The biggest advantage of KromHC by comparison is that it uses fewer parameters. Does this indicate that the performance improvement brought by using a more accurate doubly stochastic matrices structure is limited?
>
> **Response**: As noted in the original mHC paper [1], the main purpose of doubly stochastic matrices is to prevent training instabilities, which can be very costly for large-scale training. Exact doubly stochastic matrices in mHC-lite and KromHC guarantee that training instabilities cannot arise from the learnt residual matrices.
>
> The parameter efficiency of KromHC also leads to system efficiency gains. We report below the system metrics at 12 transformer blocks and $n=4$ with 8 RTX PRO 6000 GPUs. KromHC is more computationally efficient than mHC and mHC-lite in terms of **higher** throughput (663,730 tokens/sec), **lower** wall-clock per step (0.794 second) and a **lower** GPU memory footprint (68.47 GB per GPU).
>
> |Method|Throughput (tokens/sec) $\uparrow$|Wall-clock per step (second) $\downarrow$|Memory on Each GPU (GB) $\downarrow$|
> |-|-|-|-|
> |mHC|556,689|0.948|69.27|
> |mHC-lite|608,973|0.864|68.55|
> |**KromHC (Ours)**|**663,730**|**0.794**|**68.47**|
>
> Although the Kronecker structure spans a subset of the Birkhoff polytope, we find empirically that this constraint can act as a beneficial regularizer. We compared the more constrained $2 \times 2 \times 2$ against a $4 \times 2$ factorization at $n=8$ with 12 transformer blocks. The more constrained $2 \times 2 \times 2$ achieved better train loss and validation BPB.
>
> |Factorization|Train loss $\downarrow$|Validation BPB $\downarrow$|
> |-|-|-|
> |mHC-lite $n=8$|Model Size Explosion|Model Size Explosion|
> |KromHC $4 \times 2$|2.961|0.863|
> |KromHC $2 \times 2 \times 2$| **2.959**|**0.860**|
>
> [1] Xie, Z., et al. mHC: Manifold-Constrained Hyper-Connections.
>
> ---
>
> We hope these answers have addressed your concerns and would like to kindly ask you to consider raising the score. Please let us know if any questions remain, we are happy to address them.

---

> > ### Author Rebuttal · Reviewer_E9S1 · 2026-04-03
> >
> > The authors have partially addressed my concerns and provided additional experiments. I appreciate their efforts and am willing to raise my score to 4. However, I still have concerns about the method’s generalizability. For example, for a prime $n$, how should KromHC be applied in practice to construct experiments? And for a composite integer $n$ with multiple possible factorizations, which factorization is optimal for KromHC? For instance, when $n=8$, both $4\times 2$ and $2\times 2\times 2$ are possible.

---

> > > ### Author Response · Authors · 2026-04-05
> > >
> > > Thank you very much for the insightful discussion and raising your score. Please find our responses to your questions below.
> > >
> > > ---
> > >
> > > > For example, for a prime $n$, how should KromHC be applied in practice to construct experiments?
> > >
> > > **Response:** We acknowledge that $n$ is a hyper-parameter for the residual width in hyper-connections including KromHC. As modern GPUs benefit from highly factorizable numbers (e.g., 256, 1024, 2048) for optimal tensor core utilization, large prime numbers are rarely selected as hyper-parameters. While it is possible to use KromHC with a prime $n$ (as it generalizes mHC-lite), this scenario rarely occurs due to these GPU designs.
> > >
> > > ---
> > >
> > > > For a composite integer  $n$ with multiple possible factorizations, which factorization is optimal for KromHC? For instance, when $n=8$, both $ 4 \times 2$ and $2 \times 2 \times 2$ are possible.
> > >
> > > **Response:** When multiple factorizations of $n$ are possible, prime factorization (e.g., factorizing into all 2s or 3s) always yields the best parameter efficiency in KromHC (See Remark 4.5) and can act as a beneficial regularizer, making it a great default choice.
> > >
> > > ---
> > >
> > > We hope these answers have addressed your concerns. Please let us know if any questions remain, we are happy to address them.

---

### Official Review · Reviewer_JfNa · 2026-03-12

**Soundness:** 3
**Presentation:** 3
**Significance:** 3
**Originality:** 3
**Overall Recommendation:** 4
**Confidence:** 3

**Summary:**

In modern neural network architectures, residual functions acting directly on the input have proven effective in stabilizing optimization and improving the training of deep models. More recently, Hyper-Connections have emerged as an alternative in which the residual width is expanded across layers, yielding encouraging empirical gains. The main difficulty, however, is that as models scale, this widened residual space can compromise the identity-map property, which in turn may reduce training stability.

To address this issue, previous work has proposed projecting the residual connection space onto a manifold so as to preserve an identity-like mapping. The limitation of existing approaches is that they offer an unfavorable tradeoff: approximate projections remain computationally feasible but do not exactly enforce the constraint, whereas exact projections tend to incur prohibitive parameter growth as the residual width increases.

In this manuscript, the authors present an exact manifold projection scheme for Hyper-Connections that remains computationally feasible at scale. The proposed construction exploits Kronecker-product structure together with Tucker tensor decomposition in order to control parameter growth while preserving exactness. The empirical results reported in the manuscript indicate improved training stability and better performance than previous constrained Hyper-Connection variants on benchmark experiments.

**Compliance With Llm Reviewing Policy:**

Affirmed.

**Final Justification:**

The manuscript presents an original approach to a timely problem. I consider it suitable for publication.

**Key Questions For Authors:**

- How does computing the Kronecker product affect the FLOPs (or wall-clock time) during training and inference? The Kronecker product can be costly to compute, leading to higher cost or training times.

- The algorithmic justification is framed around improving stability and convergence for LLMs. I recognize that full-scale pretraining can be prohibitively expensive; however, the standard NanoChat setup has 561M parameters (D = 26). Could you also provide experiments in this model configuration? Given the reported hardware (8× NVIDIA RTX 6000), this seems feasible and would strengthen the empirical support for the claims.

**Limitations:**

It would be helpful to better understand the impact of the potentially high computational cost associated with Kronecker products, since it seems important to clarify how this design choice affects training and inference time, as well as the overall computational cost.

**Strengths And Weaknesses:**

Regarding Soundness:

The manuscript appears technically sound. The use of Kronecker products and Tucker decomposition is well established in the literature, and the mathematical formulations are presented in a clear and concise manner.

I do, however, have one question regarding the experimental evaluation. The reduction in trainable parameters and the exactness of the proposed projection are both clear advantages of the method. At the same time, operations based on Kronecker products may be computationally demanding, with the cost depending directly on the factorization of $n$ and on the resulting value of $K$, which, according to the manuscript, is preferably taken in lower dimensions, that is, with larger $K$. Since the manuscript does not discuss this potential computational drawback, it would be valuable to include empirical evidence on the computational cost of the proposed layer operations, and in particular of the computation of $H^{\mathrm{res}}$.

Regarding Presentation:

The presentation is clear and direct. The authors introduce the topic and the main contributions in a well-structured and reader-friendly manner, and the diagrams are useful and appear to be carefully designed.

The manuscript also gives a convincing motivation for the need of a manifold-constrained Hyper-Connection that controls the growth of the residual width while preserving exactness. My only reservation concerns the motivation for the use of the Kronecker product in this setting. While the authors do not claim originality for Theorem 4.2, the manuscript does not provide references for the use of Kronecker products in connection with polystochastic matrices. In particular, the fact that polystochastic matrices are closed under Kronecker products is known (e.g., Theorem 2 in https://arxiv.org/abs/2303.17278), and citing such results—along with prior work on Kronecker products in the context of the Birkhoff polytope—would improve the clarity and motivation of the proposed approach.

Regarding Significance:

There have been several recent publications on this topic at major scientific conferences, reflecting the growing interest in improving the convergence and stability of large language models. Accordingly, the topic addressed in the manuscript has received considerable attention in recent times.

Regarding Originality:

I consider the approach to be original. Although the underlying theory and properties are not new, their application here is novel and, in relevant aspects, improves upon existing solutions to this recently studied problem. However, as noted above, the paper would benefit from additional references—particularly prior work on Kronecker products in the Birkhoff polytope and related Birkhoff-type material.

---

> ### Author Rebuttal · Authors · 2026-03-28
>
> Thank you very much for the positive assessment of our work's soundness, presentation, significance and originality. Please find below our answers to your comments and questions regarding empirical computational costs, reference, and scalability.
>
> ---
>
> > **Q1**. Operations based on Kronecker products may be computationally demanding... Since the manuscript does not discuss this potential computational drawback, it would be valuable to include empirical evidence on the computational cost of the proposed layer operations... How does computing the Kronecker product affect the FLOPs (or wall-clock time) during training and inference? The Kronecker product can be costly to compute, leading to higher cost or training times.
>
> **Response**: Thank you for this suggestion which further strengthens the efficiency of the proposed KromHC. In practice, the PyTorch-native KromHC is **more efficient** than mHC and mHC-lite on system metrics such as throughput, wall-clock per step, and memory footprint. We report below the system efficiency at 12 transformer blocks and $n=4$ with 8 RTX PRO 6000 GPUs. KromHC is more computationally efficient than mHC and mHC-lite in terms of **higher** throughput (663,730 tokens/sec), **lower** wall-clock per step (0.794 second) and a **lower** GPU memory footprint (68.47 GB per GPU).
>
> |Method|Throughput (tokens/sec) $\uparrow$|Wall-clock per step (second) $\downarrow$|Memory on Each GPU (GB) $\downarrow$|
> |-|-|-|-|
> |mHC|556,689|0.948|69.27|
> |mHC-lite|608,973|0.864|68.55|
> |**KromHC (Ours)**|**663,730**|**0.794**|**68.47**|
>
> This confirms that the parameter savings in KromHC yield actual training speedups.
>
> ---
>
> > **Additional Reference**. My only reservation concerns the motivation for the use of the Kronecker product in this setting. While the authors do not claim originality for Theorem 4.2, the manuscript does not provide references for the use of Kronecker products in connection with polystochastic matrices. In particular, the fact that polystochastic matrices are closed under Kronecker products is known (e.g., Theorem 2 in [https://arxiv.org/abs/2303.17278](https://arxiv.org/abs/2303.17278)), and citing such results—along with prior work on Kronecker products in the context of the Birkhoff polytope—would improve the clarity and motivation of the proposed approach.
>
> **Response**: Thank you very much for pointing us to this valuable work. In the revised camera-ready version, we will update Theorem 4.2 to explicitly cite Theorem 2 from [https://arxiv.org/abs/2303.17278](https://arxiv.org/abs/2303.17278).
>
> ---
>
> > **Q2**. The algorithmic justification is framed around improving stability and convergence for LLMs. I recognize that full-scale pretraining can be prohibitively expensive; however, the standard NanoChat setup has 561M parameters. Could you also provide experiments in this model configuration? Given the reported hardware (8× NVIDIA RTX 6000), this seems feasible and would strengthen the empirical support for the claims.
>
> **Response**: Thank you for your suggestion to highlight the scalability of our KromHC. We have further scaled the NanoChat model to **~1.08 billion parameters** (26 layers of transformer blocks and Tokens:Params ratio of ~7). Comparing to HC, mHC, and mHC-lite at n=4, KromHC achieves the **best** train loss (2.539) and validation BPB (0.751) and the *second-best* CORE score (0.260) while being the **most efficient** in system metrics such as throughput (150,649 tokens/sec), wall-clock per step (3.499 second per step), and GPU memory footprint (86.91GB per GPU). This result highlights the scalability of the proposed KromHC.
>
> |Method|CORE score $\uparrow$|Train loss $\downarrow$|Validation BPB $\downarrow$|Throughput (tokens/sec) $\uparrow$|Wall-clock per step (second) $\downarrow$|Memory on Each GPU (GB) $\downarrow$|
> |-|-|-|-|-|-|-|
> |HC|0.240|2.554|0.756|133,726|3.932|*87.09*|
> |mHC|**0.262**|**2.539**|**0.751**|137,521|3.883|87.62|
> |mHC-lite|0.250|2.551|0.755|*138,361*|*3.807*|87.20|
> |**KromHC (Ours)**|*0.260*|**2.539**|**0.751**|**150,649**|**3.499**|**86.91**|
>
> ---
>
> We believe we have answered all your questions. We hope that you will consider increasing your score. If not, please let us know what questions remain and we would be happy to address them.

---

> > ### Author Rebuttal · Reviewer_JfNa · 2026-04-03
> >
> > Thank you for your answers. I acknowledge the scaled experiments and the new throughput statistics. I recommend discussing these in the experiments section. One little detail, in the new experiments, can I see the paramter difference \delta(K) as in Table 3?

---

> > > ### Author Response · Authors · 2026-04-05
> > >
> > > Thank you very much for the constructive suggestions. We will ensure that these results are included in the camera-ready version.
> > >
> > > ---
> > >
> > > >  In the new experiments, can I see the parameter difference $\Delta Params \ (K)$ as in Table 3?
> > >
> > > **Response:** Thank you for your suggestion to highlight the parameter efficiency of KromHC. The $\Delta Params \ (K)$ of different methods in the 1.08 billion parameter configuration is reported below. KromHC requires the **least** amount of additional parameters compared to other methods. Since the true difference between HC and mHC is the parameter-free Sinkhorn-Knopp algorithm, they share the same number of parameters. As the base model scales beyond a billion parameters, KromHC consistently has the least amount of additional parameters (about **52%** and **39%** of the $\Delta Params \ (K)$ in mHC and mHC-lite respectively) while achieving the best system metrics, train loss, and validation BPB, confirming the parameter efficiency of KromHC.
> > >
> > > | Method | $\Delta Params \ (K)$ $\downarrow$ |
> > > | --- | --- |
> > > | mHC | 8,654 |
> > > | mHC-lite | 11,423 |
> > > | **KromHC (Ours)** | **4,500** |
> > >
> > > ---
> > >
> > > We hope that these answers have addressed your concern. If any questions remain, we are happy to address them.

---

### Official Review · Reviewer_pV4F · 2026-03-14

**Soundness:** 3
**Presentation:** 3
**Significance:** 2
**Originality:** 3
**Overall Recommendation:** 4
**Confidence:** 3

**Summary:**

This paper proposes KromHC, a manifold-constrained hyper-connection variant that parameterizes the residual matrix as a Kronecker product of smaller doubly stochastic matrices. The key idea is to preserve exact double stochasticity, unlike Sinkhorn-based mHC, while avoiding the factorial parameter growth of mHC-lite. The method is motivated through a tensorized residual-stream view and evaluated on small-scale LLM pretraining built on Nanochat. Empirically, KromHC is competitive with or better than prior mHC variants on CORE and several downstream tasks while using fewer additional parameters.

**Compliance With Llm Reviewing Policy:**

Affirmed.

**Final Justification:**

My concerns have been adequately addressed, so I raised my score.

**Key Questions For Authors:**

- Can you report end-to-end runtime metrics against mHC and mHC-lite, including throughput, wall-clock per step, and memory footprint?

- How much expressive power is lost by restricting $H^{res}$ to a Kronecker-structured subset of doubly stochastic matrices, especially in the all-2x2 factorization? If this restriction is empirically benign, that would strengthen the main claim considerably.

- Why is the original HC baseline missing from the main comparison? Including it would help separate the value of manifold constraints from the value of this specific parameterization.

**Limitations:**

No. The current limitations discussion mentions unfavorable prime factorizations of n, but does not discuss the expressivity restriction induced by the Kronecker structure

**Strengths And Weaknesses:**

Strengths:

-The paper targets a real issue in hyper-connections: the tension between stability constraints, exact doubly stochastic residual mixing, and parameter scalability.

- The construction is clean and technically easy to follow. Using Kronecker products of small doubly stochastic factors is a simple but sensible way to guarantee exact double stochasticity.

- The empirical results are reasonably consistent. KromHC is usually at least competitive with mHC / mHC-lite and shows a clear parameter-count advantage.

Weaknesses:

- The paper does not analyze the expressivity cost of the Kronecker structure. In the preferred small-factor setting (e.g., all factors of size 2), the reachable residual matrices form a highly restricted subset of the Birkhoff polytope, so the paper does not really show that the exactness-efficiency trade-off is resolved without giving up flexibility.

- The practical-efficiency claim is under-supported. The paper reports parameter counts, but not wall-clock, throughput, memory, or the actual overhead of tensorization / Kronecker structure. “PyTorch-native” and “fewer parameters” do not automatically imply better end-to-end efficiency.

- The evaluation is still narrow for the scope of the claims: only two relatively small model scales, one training recipe, mostly  n=4 in the main comparisons. Some of the gains are also modest.

- The ablation section is limited. The paper does not study different factorizations of n, and does not compare against the original HC baseline

---

> ### Author Rebuttal · Authors · 2026-03-28
>
> Thank you very much for the insightful comments. We have included new experimental results to address your concerns regarding expressivity, system efficiency, and scalability.
>
> ---
>
> > **W1, first half of W4 & Q2**. The reachable residual matrices form a restricted subset of the Birkhoff polytope... The paper does not study different factorizations of n... How much expressive power is lost by restricting... If this restriction is empirically benign, that would strengthen the main claim considerably.
>
> **Response**: Thank you for this comment which allows us to strengthen the main claim. Although the Kronecker structure spans a restricted subset of the Birkhoff polytope, we observe empirically that this constraint acts as a **beneficial regularizer** to the model. Additionally, although mHC-lite spans the whole Birkhoff polytope, it exhibits **factorial** parameter growth with respect to n. For example, mHC-lite with 12 transformer blocks and $n=8$ results in ~6 billion parameters, causing the hyper-connection parameters to account for ~**97%** of the total parameters. Resolving this parameter explosion is a primary contribution of KromHC.
>
> To empirically show that the Kronecker-structure does not degrade performance, we conducted an ablation study at $n=8$ with 12 transformer blocks and different factorizations, comparing the more restricted $2 \times 2 \times 2$  against a $4 \times 2$ factorization scheme. $2 \times 2 \times 2$ achieves both better train loss and validation BPB. This confirms that the exactness-efficiency trade-off is resolved in KromHC without sacrificing model performance.
>
> |Factorization|Train loss $\downarrow$|Validation BPB $\downarrow$|
> |-|-|-|
> |mHC-lite $n=8$|Model Size Explosion|Model Size Explosion|
> |KromHC $4 \times 2$|2.961|0.863|
> |KromHC $2 \times 2 \times 2$| **2.959**|**0.860**|
>
> ---
>
> > **W2 & Q1**. The practical-efficiency claim is under-supported. The paper reports parameter counts, but not wall-clock, throughput, memory, or the actual overhead of tensorization / Kronecker structure against mHC and mHC-lite.
>
> **Response**: Thank you for this insightful suggestion. We report below the system efficiency at 12 transformer blocks and $n=4$ with 8 RTX PRO 6000 GPUs.
>
> KromHC is **more computationally efficient** than mHC and mHC-lite in terms of **higher** throughput (663,730 tokens/sec), **lower** wall-clock per step (0.794 second) and a **lower** GPU memory footprint (68.47 GB per GPU).
>
> |Method|Throughput (tokens/sec) $\uparrow$|Wall-clock per step (second) $\downarrow$|Memory on Each GPU (GB) $\downarrow$|
> |-|-|-|-|
> |mHC|556,689|0.948|69.27|
> |mHC-lite|608,973|0.864|68.55|
> |**KromHC (Ours)**|**663,730**|**0.794**|**68.47**|
>
> ---
>
> > **W3**. The evaluation is still narrow for the scope of the claims: only two relatively small model scales, one training recipe, mostly n=4 in the main comparisons.
>
> **Response**: We chose $n=4$ to allow for comparison against other methods such as mHC-lite, as mHC-lite quickly becomes infeasible as $n$ increases beyond $6$. Section 5.4 and Figure 4 show that the performance of KromHC scales as $n$ increases. KromHC is the only method that achieves exact double stochasticity and does not encounter an explosion in parameters count.
>
> In light of your comments, we have further scaled the NanoChat model to **~1.08 billion parameters** (26 layers of transformer blocks and Tokens:Params ratio of ~7). Comparing to HC, mHC, and mHC-lite at n=4, KromHC achieves the **best** train loss (2.539) and validation BPB (0.751) and the *second-best* CORE score (0.260) while being the **most efficient** in all three system metrics. This result highlights the scalability of the proposed KromHC.
>
> |Method|CORE score $\uparrow$|Train loss $\downarrow$|Validation BPB $\downarrow$|Throughput (tokens/sec) $\uparrow$|Wall-clock per step (second) $\downarrow$|Memory on Each GPU (GB) $\downarrow$|
> |-|-|-|-|-|-|-|
> |HC|0.240|2.554|0.756|133,726|3.932|*87.09*|
> |mHC|**0.262**|**2.539**|**0.751**|137,521|3.883|87.62|
> |mHC-lite|0.250|2.551|0.755|*138,361*|*3.807*|87.20|
> |**KromHC (Ours)**|*0.260*|**2.539**|**0.751**|**150,649**|**3.499**|**86.91**|
>
> ---
>
> > **Second half of W4 and Q3**. Why is the original HC baseline missing from the main comparison? Including it would help separate the value of manifold constraints from the value of this specific parameterization
>
> **Response**: [1] and [2] show that the original HC has stability issues in large-scale training due to its unconstrained nature. The performance of HC is reported in the response above and is worse than manifold-constrained HCs.
>
> [1] Xie, Z., et al. mHC: Manifold-Constrained Hyper-Connections
>
> [2] Yang, Y. and Gao, J. mHC-lite: You Don’t Need 20 Sinkhorn-Knopp Iterations
>
> ---
>
> We believe we have answered all your questions and concerns. We hope that you will consider increasing your score. Please let us know if any questions remain and we are happy to address them.

---

> > ### Author Rebuttal · Reviewer_pV4F · 2026-04-05
> >
> > Thanks for the response, my concerns have been adequately addressed.

---

> > > ### Author Response · Authors · 2026-04-05
> > >
> > > Thank you very much for the continued engagement and raising your score. We are glad that your concerns have been addressed and appreciate the insightful discussion.

---

### Decision · Program_Chairs · 2026-04-30

**Decision:**

Accept (regular)

**Comment:**

The paper introduces KromHC, a parameterization for manifold-constrained hyper-connections that fixes the scaling bottlenecks of older methods like mHC and mHC-lite. The core idea is clean and technically sound. It uses  the residual mixing matrix as a Kronecker product of smaller doubly stochastic factors (parameterized as convex combinations of permutation matrices). This preserves the stability of manifold constraints while vastly improving parameter efficiency.

The initial reviews were largely positive, though they raised fair concerns about the restricted expressivity of the Kronecker structure and the lack of large-scale, system-level efficiency data. The author responses addressed many of them, providing much stronger runtime and memory metrics, larger-scale experiments, and a solid breakdown of their factorization choices.

While this paper is more of a very smart structural engineering trick than a massive theoretical leap, it’s a highly practical solution to a real problem in an active area.